# Divergent prebiotic synthesis of pyrimidine and 8-oxo-purine ribonucleotides

Shaun Stairs[1,*], Arif Nikmal[1,*], Dejan-Krešimir Bučar[1], Shao-Liang Zheng[2], Jack W. Szostak[2,3] & Matthew W. Powner[1]

Understanding prebiotic nucleotide synthesis is a long standing challenge thought to be essential to elucidating the origins of life on Earth. Recently, remarkable progress has been made, but to date all proposed syntheses account separately for the pyrimidine and purine ribonucleotides; no divergent synthesis from common precursors has been proposed. Moreover, the prebiotic syntheses of pyrimidine and purine nucleotides that have been demonstrated operate under mutually incompatible conditions. Here, we tackle this mutual incompatibility by recognizing that the 8-oxo-purines share an underlying generational parity with the pyrimidine nucleotides. We present a divergent synthesis of pyrimidine and 8-oxo-purine nucleotides starting from a common prebiotic precursor that yields the β-ribo-stereochemistry found in the sugar phosphate backbone of biological nucleic acids. The generational relationship between pyrimidine and 8-oxo-purine nucleotides suggests that 8-oxo-purine ribonucleotides may have played a key role in primordial nucleic acids prior to the emergence of the canonical nucleotides of biology.

[1] Department of Chemistry, University College London, 20 Gordon Street, London WC1H 0AJ, UK. [2] Department of Chemistry and Chemical Biology, Harvard University, 12 Oxford Street, Cambridge, Massachusetts 02138, USA. [3] Department of Molecular Biology and Center for Computational and Integrative Biology, Howard Hughes Medical Institute, Massachusetts General Hospital, 185 Cambridge Street, Boston, Massachusetts 02114, USA. * These authors contributed equally to this work. Correspondence and requests for materials should be addressed to M.W.P. (email: matthew.powner@ucl.ac.uk).

Ribonucleotide synthesis is a key unmet requirement upon which the origin of life on Earth is contingent[1–3]. Recent progress towards prebiotic ribonucleotide synthesis uniformly fails to account for simultaneous access to both pyrimidine[4–6] and purine[7–11] nucleotides, both of which are required to generate RNA. Despite purine nucleobase synthesis by hydrogen cyanide oligomerization being first reported more than 60 years ago[3,9–11], purine nucleobases and adenosine nucleotides are only accessible in low yields and with poor regio- and stereoselectivity[7,11]. While, a prebiotically plausible synthesis of both pyrimidine ribonucleotides has been established[4–6], the glycosidations of adenine and hypoxanthine generate a complex mixture of nucleoside-like products, and there is no literature precedent for the direct glycosidation of guanine to furnish guanosine (G)[3,11]. A notable proposed pathway to the purine nucleotides, that achieves excellent N9-selectivity during ribosylation[7], remains problematic because of an unselective ribosylation (that furnished a mixture of natural furanosyl and non-natural pyranosyl isomers, and a mixture of natural β-anomers and non-natural α-anomers). Furthermore, this strategy, a variant of the classic Traube purine synthesis[8], generates a wide spectrum of glycone isomers, homologues and anomers alongside a low yield of natural nucleosides from prebiotically plausible sugar mixtures[7]. Therefore, in an inversion to the status quo of more than 50 years, a selective prebiotically plausible purine synthesis is now the most significant problem associated with prebiotic assembly of the RNA monomers.

Remarkably, all proposed prebiotic syntheses of pyrimidine and purine nucleotides have yielded either pyrimidines or purines separately, but never (yet) both by the same strategy[4–7,9–12]; while purine nucleotides can be synthesized by direct ribosylation, pyrimidine nucleotides cannot and though a complete stepwise pyrimidine nucleotide synthesis has been demonstrated (even from complex sugar mixtures[5]) a comparable strategy for purine synthesis remains elusive[4–6]. Constitutional analysis of purine and pyrimidine ribonucleotides suggests that nucleobase construction on a preformed sugar moiety would provide the simplest strategy for divergent monomer synthesis (Fig. 1). Given the lack of specificity observed during direct glycosidation of purine nucleobases, we have previously suggested that a tethered purine synthesis would overcome the limitation of intramolecular glycosidation[13,14]. Although the optimal tethering-site to direct stereo- and regiospecific purination is not clear; it is of particular note that the canonical purine nucleotides, unlike the pyrimidine nucleotides, display significant variation in their oxidation level. Nucleosides A and G are both universal and essential to biology, but interestingly they have different oxidation levels. We therefore hypothesized that the inherent flexibility in purine oxidation level may shed light upon the ideal prebiotic tether to allow divergent purine and pyrimidine synthesis. Purines particularly have a high propensity for C8-oxidation and 8-oxo-guanine (OG), for example, which though commonly mutagenic because it may give rise to G:C to T:A transversions via DNA Hoogsteen OG:A base pairs, is remarkably well tolerated during information transfer in Watson–Crick OG:C base pairs[15–21]. Moreover, *E. coli* DNA polymerase I and *Taq* DNA polymerase have both been observed to exclusively insert a correct thymine (T) nucleotide when reading through 8-oxo-adenine (OA) incorporated into gene fragments[22]. Importantly, with respect to analysis of selective purine nucleotide assembly, C8-oxidation appears to provide an ideally positioned tether for prebiotic purine nucleobase assembly by an approach that is unified with prebiotic pyrimidine synthesis. Here, we demonstrate a divergent, prebiotically plausible reaction strategy for the synthesis of both pyrimidine and purine ribonucleotides (3) on a single oxazoline scaffold.

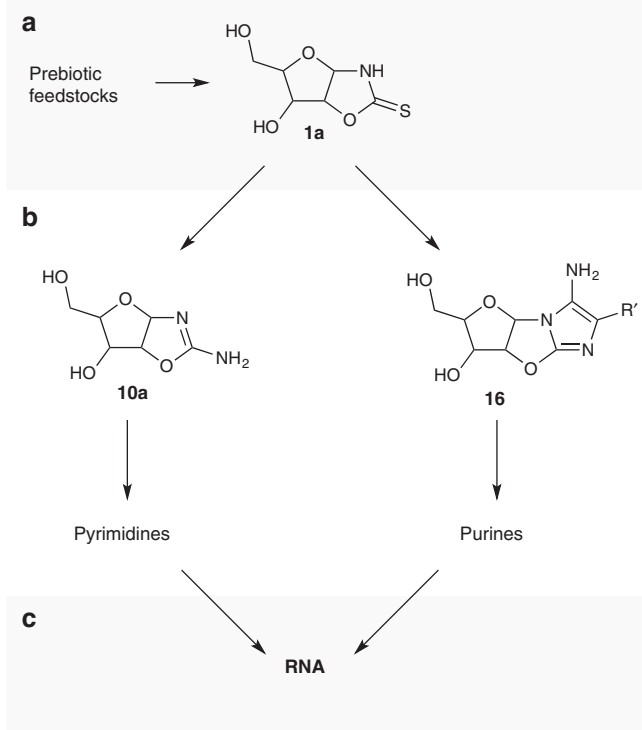

**Figure 1 | Proposed prebiotic ribonucleotide synthesis by divergent nucleobase assembly on a universal furanosyl-oxazoline scaffold.**
(**a**) Unitary phase: the reaction of prebiotically available feedstock molecules to assemble a universal sugar scaffold. (**b**) Divergent phase: single sugar scaffold diverges through congruent reactions to yield both pyrimidine and purine precursors with controlled furanosyl sugar structure, pentose selectivity and stereocontrol. R′ = CN or CONH₂. (**c**) Convergent phase: concomitantly synthesized purine and pyrimidine nucleotides are collectively assembled to yield polymeric RNAs.

## Results

**Overview**. We envisaged that a purine-8,2′-anhydronucleoside linkage could be exploited to deliver regio- and stereospecific glycosidation, with direct generational parity to prebiotic pyrimidine synthesis via a pyrimidine-2,2′-anhydronucleoside intermediate (Fig. 2)[4]. We hypothesized that positioning sulfur at the C2-carbon atom of an oxazolidinone thione (1), followed by chemoselective sulfide-activation and amine displacement, would provide the chemical differentiation required for divergent pyrimidine (amine = ammonia, **2a**) and 8-oxo-purine (amine = HCN-trimer, **2b–c**) nucleotide synthesis from one common precursor. A common thione precursor would bypass the unstable free sugars and problematic direct purine nucleobase glycosidation, and move us a step closer to understanding the unified origins of purine and pyrimidine nucleotides in biology.

**Furanosyl-selective sugar synthesis**. Oxazole **4a** is a key ribonucleotide precursor which derives from glycolaldehyde (**5**) and cyanamide (**6**) (Fig. 2)[4]. The C2-carbon atom of **4a** is regiospecifically positioned as the C2-pyrimidine carbon atom in abiotic pyrimidine synthesis upon reaction with glyceraldehyde (**7**) and cyanoacetylene (**8**). To introduce a sulfur atom at the C2-carbon atom of oxazole **4b**, and consequently acquire the plasticity needed to diverge towards both purine and pyrimidine nucleotides, we reasoned that synthesis should commence with

**Figure 2 | Divergent ribonucleotide synthesis.** Prebiotic assembly of cytidine-2′,3′-cyclic phosphate **3C**, uridine-2′,3′-cyclic phosphate **3U**, 8-oxo-adenosine-2′,3′-cyclic phosphate **3OA** and 8-oxo-inosine-2′,3′-cyclic phosphate **3OI**. Glycolaldehyde (**5**) reacts with cyanamide (**6**) and thiocyanic acid (**9**) to furnish 2-aminooxazole **4a** and 2-thiooxazole **4b**, respectively. 2-Aminooxazole **4a** is a known prebiotic precursor of pyrimidine nucleotides (**3C** and **3U**) and 2-thiooxazole **4b** is demonstrated to be a precursor of both pyrimidine nucleotides (**3C** and **3U**) and 8-oxo-purine nucleotides (**3OA** and **3OI**). Oxazole **4b** undergoes reaction with glyceraldehyde (**7**) to yield thione **1a**. Chemical activation of thione **1a** provides a second point of divergence (thione **1b**), which yields pyrimidine precursor **10a** upon reaction with ammonia (**2a**) or purine precursors **16b** or **16c** upon reaction with hydrogen cyanide oligomers (**2b** or **2c**, respectively). Cyanovinylation and formylation of **10a** and **16b/c** and subsequent urea-mediated phosphorylation leads to the congruent synthesis of pyrimidine nucleotides (**3C** and **3U**) and 8-oxo-purine nucleotides (**3OA** and **3OI**). The two points of chemical divergence, Divergence A and divergence B, are marked with green boxes and the chemical convergence, Convergence, on nucleotide monomers is marked with an orange box.

prebiotically plausible thiocyanic acid (**9**)[23–25], which can be generated quantitatively from hydrogen cyanide and sulfur[26]. Treatment of glycolaldehyde (**5**; 0.1 M) with aqueous thiocyanic acid (**9**; 0.5 M), yields 2-thiooxazole (**4b**) in 85% yield (Fig. 2 and Supplementary Fig. 1), and it is also of note that **4b** can purified by crystallization directly from water and can be transported by sublimation, providing simple prebiotically plausible mechanisms for purification, accumulation and transport of 2-thiooxazole (**4b**)[27,28]. Pleasingly, we found that at near neutral pH (pH 4–9; Supplementary Table 1) the reaction of 2-thiooxazole (**4b**; 0.25 M) with glyceraldehyde (**7**; 0.5–1 M) in water at 60 °C yielded a mixture of the pentose oxazolidinone thiones (**1a**). The reaction is sluggish below pH 6, but more rapid above pH 7 yielding up to 74% thione **1a** in 24 h. The

reaction proceeds with high ribo-/arabino-diastereoselectivity (70%, ribo/arabino 1:1; 30% lyxo/xylo), which is of note because both ribo- and arabino-thiones **1** are only one stereochemical inversion from the β-ribo-stereochemistry of RNA. Furthermore, furanosyl selectivity is equally important en route to RNA, and crystallization of all the diastereomeric products of the reaction of 2-thiooxazole (**4b**) and glyceraldehyde (**7**) demonstrates that only the minor lyxose component was furnished as a mixture of furanosyl- and pyranosyl-isomers (p-lyxo-**1a**/f-lyxo-**1a**; 2:1; Supplementary Fig. 4). The ribo-, arabino- and xylo-isomers are generated with complete furanosyl selectivity. Synthesis of **1a** achieves, in two prebiotically plausible steps, our first goal: to selectively position sulfur at the C2-position of the pentose oxazolidinones.

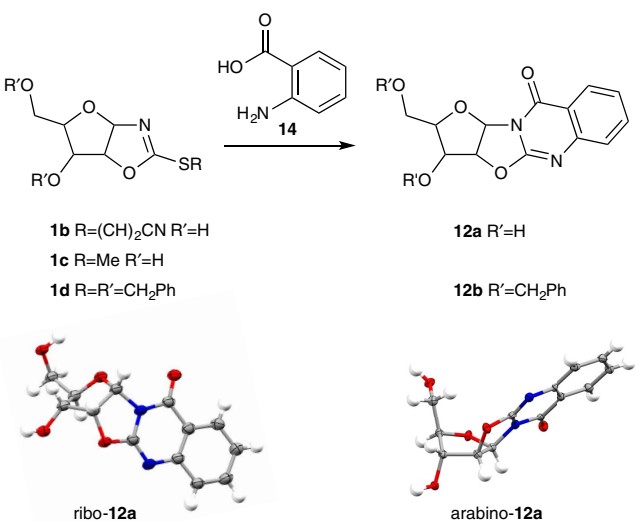

ribo-**12a**

arabino-**12a**

**Figure 3 | Model quinazolinedione synthesis.** One-pot cyanovinylation of arabinofuranosyl-oxazolidinone thione (**1a**) and quinazolinedione **12a** synthesis in water was tested as a model nucleobase synthesis strategy to investigate the pH dependence of cyanovinyl- and alkyl-sulfide displacement from *S*-cyanovinyl arabinofuranosyl-oxazolidinone thione (**1b**) and *S*-methyl arabinofuranosyl-oxazolidinone thione (**1c**), respectively. Single crystal X-ray structure of ribo-quinazolinedione (ribo-**12a**) and arabino-quinazolinedione (arabino-**12a**).

**Site-specific thione activation.** Next, we investigated the selective activation of thione **1a**. En route to the pyrimidine nucleotides, cyanovinylation of pentose aminooxazoline **10a** occurs with nearly complete selectivity at the endocyclic N1-nitrogen atom, followed by rapid annulation to yield anhydronucleoside **11** (Fig. 2). However, it was necessary to introduce stoichiometric phosphate to maintain pH <7 and prevent hydrolysis of the 2,2′-anhydronucleoside linkage that was required for C2′-stereochemical inversion to prebiotically access β-ribonucleotides[4]. Here, we envisaged cyanoacetylene (**8**) as ideally suited to activate thione **1a** in aqueous solution due the slow reaction of cyanoacetylene (**8**) with water and the excellent electron-withdrawing properties of the cyanovinyl moiety, that would activate the C2-carbon atom of **1** to nucleophilic addition[29,30]. Pleasingly, we observed quantitative cyanovinylation of thione **1a** (0.25–1 M), upon incubation with aqueous cyanoacetylene (**8**; 1.1–2 equiv.) with click-like efficiency, and complete regio- and stereocontrol to furnish *S*-cyanovinyl thione **1b** in water at room temperature in one hour (Supplementary Fig. 5). Due to the higher p$K_a$ of anhydronucleoside **11** than *S*-cyanovinyl thione **1b**[31], and sulfur-prohibited annulation, no increase in pH was observed during cyanovinylation of thione **1a**. Increasing pH is a hallmark of the addition of cyanoacetylene (**8**) to aminooxazoline (**10a**) in water, rendering pH-buffered cyanovinylation essential to pyrimidine synthesis[4], however no buffer was required to control the reaction of **1a** with cyanoacetylene (**8**). Thus, cyanoacetylene (**8**) provides a superbly controlled and quantitative activation of **1a** in water.

**Aminooxazoline synthesis.** It has been previously noted that *S*-alkyl thione **1c** is 'singularly unreactive towards nucleophiles'[32], and to our knowledge, thiolate displacement from **1c** (R = Me) by ammonia (**2a**) to yield **10a** has only been observed upon 'treatment with formamide at 90 °C for 3 h' (which was likely contaminated with ammonium formate and can slowly release

ammonia (**2a**) and formic acid)[33]. However, the *S*-benzyl thione **1d** (R = CH₂Ph) has been substituted during quinazolinedione **12** synthesis (Fig. 3)[34,35], but reports are limited to anthranilic acid derivatives in ethanol or *t*-butanol. We hypothesized that thione protonation and weak amine solvation were both essential to these limited examples in formamide or absolute alcohol solvents. In agreement with literature reports, we do not observe reaction between *S*-alkylated thione (**1c**, R = Me; p$K_{aH}$ = 2.4; see Supplementary Fig. 11) and ammonia (**2a**; p$K_{aH}$ = 9.2) in aqueous solution. However, we considered that *S*-cyanovinyl thione **1b**, due to the greatly increased electron-withdrawing effect of the cyanovinyl moiety with respect to both methyl and benzyl moieties, might take part in sulfide displacement reactions even at higher pH where substantial thione protonation would not occur. Moreover, we also predicted that judiciously chosen amines would be able to displace even alkyl thiolates at pHs where the thione is (partially) protonated and the amine is present as the free-base.

As a preliminary test of our hypotheses and to explore the prebiotic synthesis of **1c**, we investigated thiol exchange. We were pleased to observe that sequential addition of cyanoacetylene (**8**) and methanethiol to thione **1a** furnished **1c** in up to 50% yield at pH 6 (Supplementary Fig. 13), demonstrating prebiotically plausible access to **1c**. *S*-alkyl thione **1c** is more stable than *S*-cyanovinyl thione **1b**, but we predicted that the reactivity of **1c** could be controlled (switched on/off) through protonation. To test this hypothesized pH-switch, we investigated the synthesis of quinazolinedione **12** in water (Fig. 3) across a broad pH range. Interestingly, we observed near-quantitative conversion of **1c** to ribo-**12a** and arabino-**12a** in water between pH 2 and 6, supporting our hypothesis that thione protonation was essential for the activation of *S*-alkyl thiones. Moreover the reaction was severely retarded under alkaline conditions (pH >6, Supplementary Fig. 17), demonstrating that the reactivity of **1c** is readily modulated through pH-control. Pleasingly, aqueous cyanovinylation followed by *in situ* reaction with anthranilic acid (**14**) generated quinazolinedione **12a** in water at all pH's investigated (pH 3–10). It is of note that the maximal efficiency of displacement occurred at pH 6, where near-quantitative conversion of 0.25 M **1b** to quinazolinedione **12a** was observed within 6 h at room temperature.

We next investigated thione displacement with ammonia (**2a**; p$K_{aH}$ = 9.2)[25], to establish a novel route for the prebiotic assembly of pyrimidine nucleotides from the precursor **10a**. Unlike the case for simple *S*-alkyl thiones such as **1c**, aqueous ammonia (**2a**) efficiently displaces thiolate **13** from cyanovinyl thione **1b** and incubation of arabino-**1b** or ribo-**1b** (0.25 M) with ammonium chloride (1 M, pH 8.5–10.5) returns arabino- and ribo-aminooxazoline **10a** (15–23%), from their respective thiones **1b**. Interestingly, the major by-products are the precursor thione **1a** (37–42%) and a white crystalline precipitate of dicyanovinyl sulfide **15** (Fig. 4)[36]; these by-products suggested regeneration of thione **1a** results from rapid nucleophilic addition of thiolate **13** to the cyanovinyl moiety of **1b**. However, the greatly increased efficacy of cyanovinylation of **1a** suggested that **1b** could be regenerated *in situ* by reactivation of **1a** (in the presence of **10a**). Indeed, we observed that addition of cyanoacetylene (**8**; 0.25 M) to an aqueous solution of thione **1a** (0.24 M) and **10a** (0.24 M) between pH 7 and 10.5, led to chemospecific cyanovinylation of **1a**, which then reacted with ammonia to yield aminooxazoline **10a** (60%). Furthermore, we observed that repeated cyanovinylation and incubation of thione **1a** in ammonia solution yielded up to 45% arabino-aminooxazoline arabino-**10a** and 33% ribo-aminooxazoline ribo-**10a** (over two cycles of cyanovinylation at pH 10.5, without need for any intermediate steps of purification) leading to a remarkably pure solution of aminooxazoline **10a** (Supplementary Figs 18 and 19).

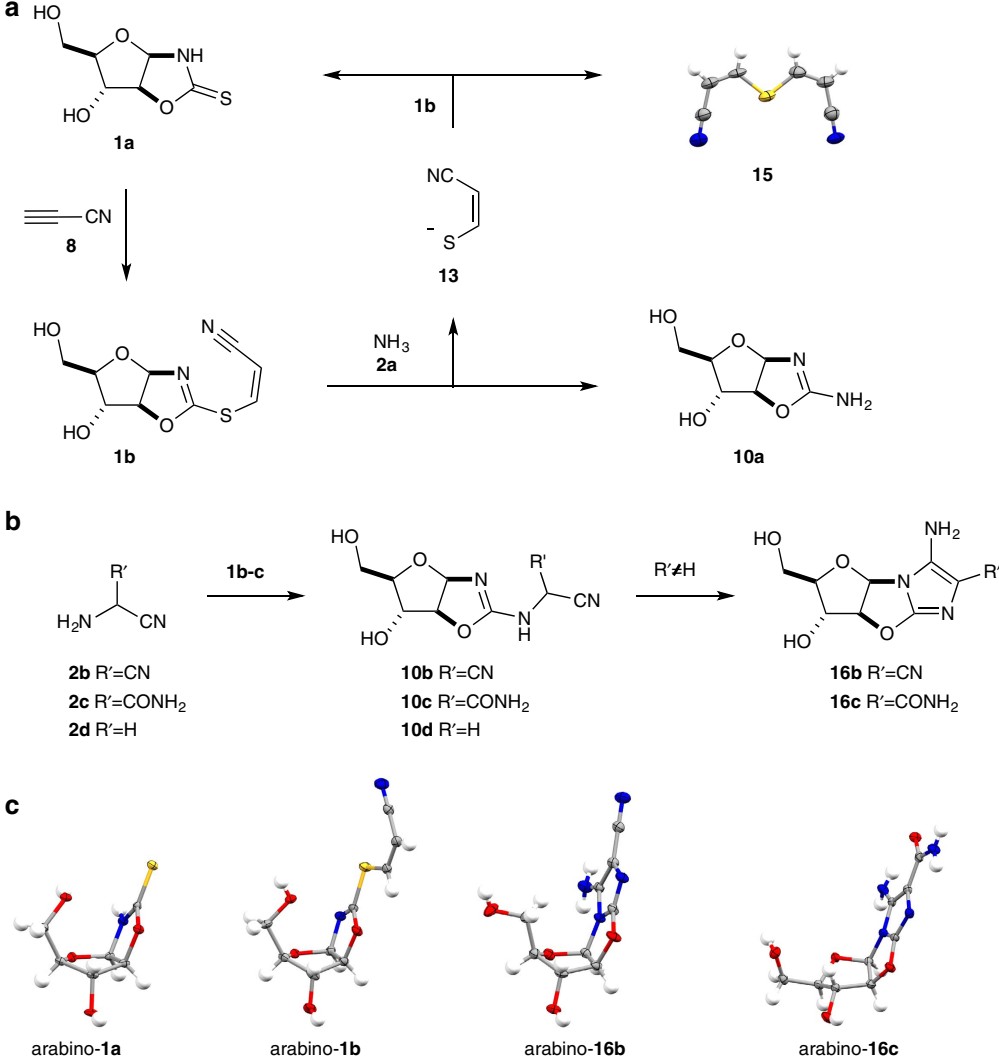

**Figure 4 | Divergent aminooxazoline synthesis by sulfide displacement.** (**a**) Ammonia (**2a**) displacement of cyanovinylthiolate **13** from cyanovinyl arabinofuranosyl-oxazolidinone thione (**1b**) to furnish pyrimidine precursor arabinofuranosyl-aminooxazoline (**10a**) and concomitantly regenerate arabinofuranosyl-oxazolidinone thione (**1a**). (**b**) Aminonitrile **2b–d** displacement to furnish purine precursors arabinofuranosyl-aminooxazoline **10b–d**, and selective cyclization of arabinofuranosyl-aminooxazoline **10b–c** to aminoimidazoles **16b** and **16c**. (**c**) Single crystal X-ray structure of dicyanovinyl sulfide (**15**), arabinofuranosyl-oxazolidinone thione (**1a**), cyanovinyl arabinofuranosyl-oxazolidinone thione (**1b**), arabinofuranosyl-aminoimidazole (**16b**) and arabinofuranosyl-aminoimidazole (**16c**).

In addition, ribo-**10a** was observed to spontaneously crystallize from the reaction mixture after two cycles of cyanovinylation and ammonolysis[5]. As arabino-**10a** (refs 4,37) and ribo-**10a** (refs 5,6) are both key intermediates en route to pyrimidine ribonucleotides this additional synthetic strategy further increases the potential of **10a** as a prebiotically plausible precursor of RNA.

**Purine elaboration**. We next investigated the synthesis of purine moieties. The oligomerization of hydrogen cyanide has widely been proposed as a key route to purine nucleobases[3,7,9–11], and guided by the recently reported synthesis of purine precursors from aminonitrile **2b**[7], we recognized that, constitutionally, **2b** and it's hydrolysis product **2c** (Figs 2 and 4)[38–42], have the ideal ambident reactivity to substitute thione **1** and then cyclize to build the imidazole moiety of purine nucleobases upon a sugar scaffold tethered by the 8,2′-anhydro-linker required for phosphorylation and C2′-stereochemical inversion (Fig. 2)[4,43].

Although the stability of trimer **2b** is limited under high-pH conditions, it is a key intermediate in the oligomerization of hydrogen cyanide and is both stable and isolable at low pH. Furthermore, **2b** can be readily converted into aminonitrile **2c**[42], which is stable across a broad pH range. Accordingly, we next incubated thione **1c** together with aminonitrile **2c** in water between pH 3 and 6, chosen as the pH range required for the quantitative synthesis of anhydrocytidine **11** from aminooxazoline **10a** during pyrimidine nucleotide synthesis[4]. Good yields of aminooxazoline **10c** (81%) were observed upon incubation of **1c** (0.25 M) and aminonitrile **2c** (0.5 M) at pH 4.5 for 8 h at room temperature. Furthermore, the product was observed to undergo facile cyclization to give key intermediate aminoimidazole **16c** in up to 59% yield (over four steps from **1a** in water without requiring purification of intermediates). It is of particular note that **16c** was observed to directly crystallize from water in these crude reaction mixtures—albeit in lower yield (13%) than can be recovered chromatographically. Direct crystallization of aminoimidazole **16c** from water is a plausible

**Figure 5 | Aminoimidazole formylation.** Incubation of aminoimidazole (**16b–c**) in formamidine/formamide solution yields anhydropurines **17A** and **17I**, respectively. Formamidine and hydrogen cyanide in formamide provide comparable yields for formylation of **16c**, whereas formamidine provides an excellent yield of **17A** from **16b** likely exploiting the electrophilicity of the nitrile moiety of **16b**.

prebiotic mechanism to purify material during sequential chemical reactions, and crystallization could allow the accumulation of reservoirs of aminoimidazole **16c** under prebiotically plausible conditions[5]. The cyclization of **10c** was observed but sluggish at pH 4–5, more rapid at pH 7–9 and highly facile at pH 11–13. It is of note that although thione **1c** has previously been reported to be unreactive to nucleophiles[32], displacement occurs readily with aminonitriles at pH 3–6, and aminooxazolines **10b–d** can all be synthesized from S-alkyl thione **1c** providing access to aminoimidazoles **16b** (15%) and **16c** (59%) and aminooxazoline **10d** (76%). It is likely that the combination of thione protonation (**1c** $pK_{aH} = 2.4$) and the remarkably low amine $pK_a$ (decreased through the inductive effect of the nitrile moiety; **2b** $pK_{aH} = 6.5$ (ref. 44), **2c** $pK_{aH} = 3.4$, **2d** $pK_{aH} = 5.6$; Supplementary Figs 8–11)[45] results in the now facile displacement of sulfide from S-alkyl thione **1c**. Furthermore, although aminonitrile **2b** is a less efficient nucleophile than aminonitrile **2c**, the dinitrile **10b** cyclizes much more rapidly than nitrile **10c**, and consequently quantitatively yields **16b** after 1 h at pD 9. The improved efficiency of dinitrile cyclization is likely due to both nitrile-nucleophile effective molarity and the increased electron withdrawal of the α-nitrile with respect to the α-amide. It is also noted that though reaction of **1c** with glycine nitrile (**2d**) yielded aminooxazoline **10d** in excellent (76%) yield, cyclization of **10d** was not observed under any condition investigated and was readily isolated as the uncyclized aminooxazoline (61%). The observed differential reactivity of purine precursors **10b** and **10c**, with respect to **10d**, demonstrates an inherent and important selectivity for the cyclization of purine precursors.

To assemble the purine heterocycle from key aminoimidazole intermediates **16b–c** a fifth carbon is required, and hydrogen cyanide derivatives again appeared to be the ideal prebiotic choice to convert aminoimidazoles **16b–c** to anhydropurines **17A** and **17I**. To test this hypothesis, we incubated **16b** and **16c** in formamide at 100 °C and observed direct conversion to **17A** (10%, 96 h) and **17I** (3%, 72 h) (Fig. 5). Pleasingly, addition of formamidine (10 equiv.) markedly improved the yield and rate of anhydro-adenosine **17A** synthesis (65%, 5 h; Fig. 5 and Supplementary Fig. 29); however, formamidine only marginally improved the yield of inosine **17I** (11%, 48 h). It is therefore of

note canonical nucleobase adenine (A) is efficiently synthesized, but wobble base-pairing inosine (I) is only ineffectually synthesized[46]. Accordingly, our results suggest that further investigation of the concomitant elaboration of aminoimidazoles **16b** and **16c** may uncover conditions leading to both anhydro-adenosine **17A** and anhydro-guanosine **17G**. Indeed, incubation of **16b** and **16c** in formamide/formamidine at 100 °C for 5 h yields abundant anhydro-adenosine **17A** (60%), but only 4% **17I** alongside 35% residual **16c**. However, we have not observed the synthesis of **17G** within the limits of detection upon incubation of **16c** with cyanate, urea or cyanogen in formamide at 100 °C.

**Phosphorylation and stereochemical inversion.** Efficient phosphorylation of pyrimidine **11** is achieved by drying an aqueous solution of **11**, urea/formamide, and inorganic phosphate[4]. Phosphorylation is selective for the 3′-OH, leading into intramolecular rearrangement of **11**-3′-phosphate to the cytidine-2′,3′-cyclic phosphate **3C** with the desired β-ribo-stereochemistry. Both kinetic and thermodynamic properties of this system control selection; $n \rightarrow \pi^{\star}$ donation suppresses nucleophilicity of the 5′-hydroxyl and monophosphate synthesis is reversible[4,47–51], whereas 2′,3′-cyclic phosphates are generated by a different mechanism and are synthesized irreversibly[3,4,48]. Interestingly, single crystal x-ray diffraction of purines **17A** and **17I** demonstrated not only that the anhydronucleoside bond had been retained in the required site to activate the anhydropurines for arabino→ribo stereochemical inversion but also that, in the solid state, an observable interaction between the 5′-hydroxyl and 8-carbon atom was displayed. It is particularly of note that there is a striking similarity between the crystal structures of anhydrocytidine **11** (refs 4,47), and anhydropurines **17A**[52] and **17I** (Fig. 6). Therefore, we next investigated the phosphorylation of anhydropurines **17A** and **17I**, finding that both were smoothly converted to 2′,3′-cyclic phosphates (55–70%); conversion to 8-oxo-adenosine **3OA** (22% + 33% 2′,3′-cyclic-5′-bisphosphate **18OA**) and 8-oxo-inosine **3OI** (38% + 32% 2′,3′-cyclic-5′-bisphosphate **18OI**) was observed under urea-mediated phosphorylation (Fig. 6 and Supplementary Table 2). These phosphorylations of **17A** and **17I** were remarkably clean; we suggest that the smooth conversion of **17A** and **17I** to ribonucleotide cyclic phosphates can, in part, be attributed to the stability of the 8,2′-anhydronucleoside linkage[53]. Whereas **11** is readily and rapidly hydrolysed in aqueous solution (pH > 6.5), and significant hydrolysis of **11** is observed under urea-mediated phosphorylation, preventing intramolecular inversion and cyclic phosphate formation, **17A** and **17I** are remarkably resistant to hydrolysis. Even upon extended incubation at elevated pH **17A** and **17I** do not undergo hydrolysis. Instead, **17A** undergoes isomerization to 8,5′-anhydronucleoside **19** (55%, pH 11, 40 °C, 24 h) and oxirane **20** (60%, pH 13, 40 °C, 2 h) rather than hydrolysis (Fig. 6 and Supplementary Fig. 34). Furthermore, due to the irreversible nature of 2′,3′-cyclic phosphate synthesis, further incubation of **3OA/18OA** (22% + 33%) with a diol, for example glycerol (10 equiv.) or cytidine (1 equiv.), leads to an increased yield of cyclic phosphate **3OA** (38% and 41%, respectively) by sequestering phosphate from the (reversible) equilibrium with the 5′-phosphate moiety of **18OA** (Supplementary Table 2), demonstrating the predisposition of cyclic phosphate synthesis during urea-mediated phosphorylation.

**Chemical divergence of purines and pyrimidines.** Given the importance of the biochemical interplay between purines and pyrimidines, we next sought to investigate the concomitant

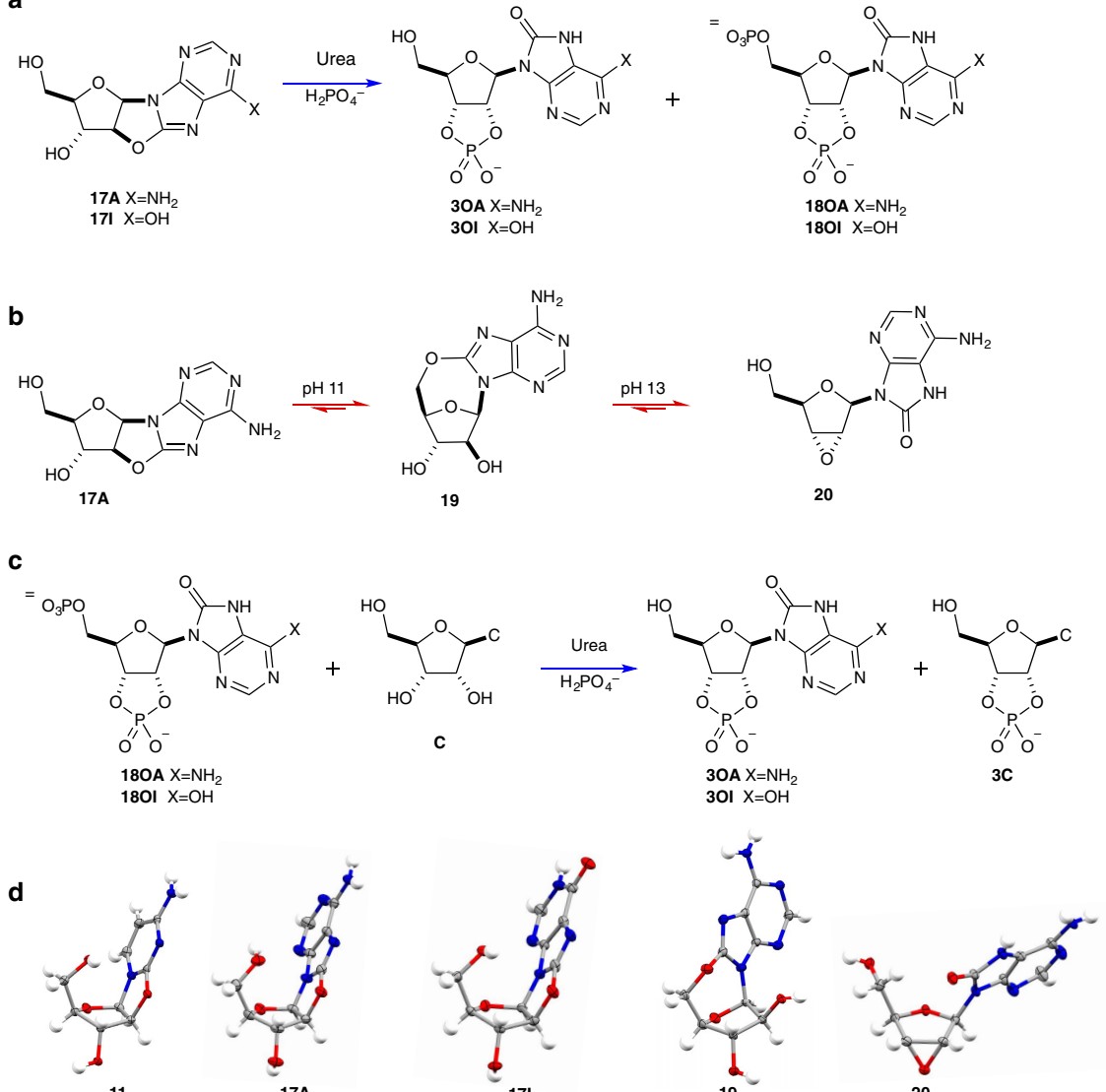

**Figure 6 | Urea-mediated phosphorylation of 8,2′-O-cyclo-purines.** (**a**) Urea-mediated phosphorylation of 8,2′-O-cyclo-adenine (**17A**) and 8,2′-O-cyclo-inosine (**17I**) to yield 8-oxo-adenosine-2′,3′-cyclic phosphate (**3OA**) and 8-oxo-inosine-2′,3′-cyclic phosphate (**3OI**). (**b**) Equilibration of 8,2′-O-cyclo-adenine (**17A**) with 8,5′-anhydro-8-oxyadenine (**19**) and 2′,3′-anhydro-8-oxo-adenosine (**20**) upon incubation in alkaline solution. (**c**) Urea-mediated conversion of bisphosphate **18** to monophosphate **3**, driven by the irreversible synthesis of 2′,3′-cyclic phosphates. (**d**) Single crystal X-ray structures of anhydrocytidine (**11**), 8,2′-O-cyclo-adenine (**17A**), 8,2′-O-cyclo-inosine (**17I**), 8,5′-anhydro-8-oxyadenine (**19**) and 2′,3′-anhydro-8-oxo-adenosine (**20**). Structure **11** was reported in ref. 4 and is shown for comparison to structures **17A** and **17I**.

synthesis of both classes of nucleotide. Interestingly, ammonia (**2a**) displacement of thiolate **13** from cyanovinyl adduct **1b** (Fig. 3) results in divergent synthesis of aminooxazoline **10a** and thione **1a** in comparable yields. This reaction seemed ideally suited to establish the chemical divergence required for the concomitant synthesis of the purine and pyrimidine ribonucleotides. Owing to the increased nucleophilicity of thione **1a** with respect to aminooxazoline **10a**, **1a** can be quantitatively and chemoselectively re-cyanovinylated in the presence of **10a** or with excess cyanoacetylene (**8**) concomitant synthesis of anhydronucleoside **11** and anhydronucleoside precursor **1b** can be achieved with remarkable efficiency (Supplementary Figs 6 and 7). Furthermore, reaction of **1b** with aminonitrile **2c** is achieved with good selectivity in the presence of either **10a** or **11**, allowing for the concomitant synthesis of pyrimidine anhydronucleosides **11** and purine anhydronucleoside precursor

**16c** through the repeated action of cyanoacetylene (**8**), ammonia (**2a**) and aminonitrile **2c** upon a single nucleotide precursor **1a** (Supplementary Figs 24–26). The direct formylation of **16c** in the presence of anhydronucleoside **11** is observed to yield **17A** (53%) alongside 45% residual **11** after 3 h (Supplementary Figs 31 and 32). Finally, we also demonstrated the simultaneous phosphorylation and intramolecular inversion of both pyrimidine and purine anhydronucleosides **11** and **17A** in comparable 44% and 34% yields, respectively, by urea-mediated phosphorylation. The remarkable parity between the reactions that yield pyrimidine and 8-oxo-purine nucleotides, and the observed tolerance for the simultaneous pyrimidine and purine synthesis, suggests that the 8-oxo-purines may have formed a bridge between prebiotic and biological information transfer, thereby facilitating access to the prebiotic synthesis of RNA.

## Discussion

We have demonstrated the generational relationship between pyrimidine and purine nucleotides by constructing both heterocycles on the same sugar scaffold. The formal (C8)-oxidation level of the 8-oxo-purine moiety allows constitutional assembly of the 8-oxo-purine ribonucleotides at the same oxidation level as the canonical pyrimidine ribonucleotides. It is also of note that glycine nitrile adduct **10d**, which is not at the correct oxidation level to yield an aminoimidazole, is not observed to undergo cyclization; cyclization is only observed with hydrogen cyanide trimers **10b** and **10c** to form the core imidazole motif of the purines. These highly selective cyclizations strongly suggest that renewed investigation of prebiotic aminonitrile **2b** is warranted[7,54], avoiding the uncontrolled high-pH oligomerization of hydrogen cyanide. The recent observation that stepwise redox coupling of cyanide and aldehydes, under mild photoredox conditions, negates the requirement for uncontrolled high-pH formaldehyde oligomerization during prebiotic sugar synthesis[25], suggests that similar redox coupling strategies could provide an alternative mechanism for hydrogen cyanide oligomerizations in a controlled and stepwise manner. We are also currently investigating the chemoselective reduction of the 8-oxo-purine ribonucleotide-2′,3′-cyclic phosphates **3OA** and **3OI** to the canonical ribonucleotides **3A** and **3I**. However, given the strongly disfavoured Hoogsteen pairing in RNA[55], it is also possible that the 8-oxo-purine ribonucleotides would allow for reasonably accurate information transfer during the non-enzymatic copying of RNA templates. Given the parity between pyrimidine and 8-oxo-purine ribonucleotide-2′,3′-cyclic phosphate **3OA** and **3OI** syntheses, **3OA** and **3OI** are good candidates for monomeric units in the early stage of replication and template-directed RNA synthesis. Moreover, 8-oxo-purines have remarkably stable glycosidic linkages[56,57], and when incorporated into RNA strands can mimic the function of a flavin in photorepair suggesting this motif could have provided an essential role in prebiotic redox and repair processes[17]. Although purine-pyrimidine Watson–Crick base pairing is thermodynamically less stable with 8-oxo-nucleotides[58], the resulting lower melting temperature of the RNA duplex may have been advantageous by facilitating thermal strand separation. Accordingly, our results suggest that further investigation of the informational and functional properties of the 8-oxo-purine ribonucleotides is warranted.

## Methods

**2-Thiooxazole (4b).** Glycolaldehyde (**5**; 1 g, 16.7 mmol) and potassium thiocyanate (3.24 g, 33.3 mmol) were dissolved in water (3 ml). The mixture was cooled to 0 °C and HCl (37%, 2.10 ml) was added drop wise. The reaction mixture was incubated for 2 h at room temperature and then at 80 °C, and NMR spectra were periodically acquired. After 24 h the starting material had been consumed and the reaction was cooled to room temperature (Supplementary Fig. 1). The organics were then extracted into ethyl acetate (3 × 50 ml), washed with brine (3 × 30 ml) and dried over MgSO$_4$. The solvent was removed under reduced pressure and the resulting solids were crystallized from CH$_2$Cl$_2$ to yield 2-thiooxazole (**4b**; 1.43 g, 14.2 mmol, 85%) as a yellow crystalline solid (Supplementary Fig. 61). M.p. 140–144 °C (Lit 147 °C (ref. 27)). IR (solid, cm$^{-1}$) 3,117 (NH), 1,587 (C=C), 1,478 (C=S). $^1$H NMR (600 MHz, D$_2$O) 7.33 (1H, d, $J = 4.5$ Hz, H4), 7.05 (1H, d, $J = 4.5$ Hz, H5). $^{13}$C NMR (151 MHz, D$_2$O) 188.1 (C2), 130 (C4), 117.2 (C5). HRMS (m/z) calculated for C$_3$H$_3$NOS [M]$^+$, 100.9930; found, 100.9930. Crystallographic and refinement parameters are shown in Supplementary Table 4.

**Pentose oxazolidinone thiones (1a).** 2-Thiooxazole (**4b**; 75.6 mg, 0.75 mmol) and 4,4-dimethyl-4-silapentane-1-sulfonic acid (DSS; 20 mg; internal standard) were dissolved in D$_2$O (1.5 ml) to give a 500 mM solution of **4b**. The solution was adjusted to pD 7 with NaOD (4 M) and 250 or 500 μl was added to glyceraldehyde* (**7**; 500 mM or 1 M). Solutions were made up to 500 μl with D$_2$O where necessary. The solutions were stirred at 60 °C for 24 h and the solution was re-adjusted to pD 7 every 6 h (if necessary). Aliquots (50 μl) were taken after 24 h, diluted with D$_2$O (450 μl) and analysed by NMR spectroscopy. The presence of **1a** was confirmed by spiking with authentic samples (Supplementary Methods). Yields were calculated

by measuring the ratio of product **1a**:DSS at 24 h (Supplementary Table 1 and Supplementary Figs 2 and 3). *2-Thiooxazole (**4b**) reacts with rac-, D- and L-glyceraldehyde (**7**) to furnish rac-, D- and L-pentose oxazolidinone thiones (**1a**), respectively.

**Ribofuranosyl oxazolidinone thione (ribo-1a).** Overall, 67% from ribose (Supplementary Methods) after 4 days at 60 °C (Supplementary Fig. 54). M.p. 172–175 °C decomp. (Lit 167–168 °C[59]). IR (Solid, cm$^{-1}$) 3,438 (NH), 3,329 (OH), 2,993, 2,920, 2,902 (CH), 1,521 (C=S). $^1$H NMR (600 MHz, D$_2$O) 5.94 (1H, d, $J = 5.4$ Hz, H1′), 5.35 (1H, $t$, $J = 5.4$ Hz, H2′), 4.26 (1H, dd, $J = 9.3$, 5.4 Hz, H3′), 3.97 (1H, ABX, $J = 12.6$, 1.9 Hz, H5′), 3.79 (1H, ddd, $J = 9.3$, 4.9, 1.9 Hz, H4′), 3.76 (1H, ABX, $J = 12.6$, 4.9 Hz, H5″). $^{13}$C NMR (151 MHz, D$_2$O) 191.1 (C2), 88.9 (C1′), 85.8 (C2′), 79.2 (C4′), 70.8 (C3′), 59.9 (C5′). HRMS (m/z) calculated for C$_6$H$_{10}$NO$_4$S [M + H$^+$]$^+$, 192.0331; found, 192.0340. Crystallographic and refinement parameters are shown in Supplementary Table 3.

**Arabinofuranosyl-oxazolidinone thione (arabino-1a).** Overall, 65% from arabinose (Supplementary Methods) after 6 days at 60 °C (Supplementary Fig. 55). M.p. 137–139 °C (Lit 132–133 °C[59]). IR (Solid, cm$^{-1}$) 3,394 (NH), 3,299 (OH), 2,991, 2,943, 2,931, 2,871 (CH), 1,482 (C=S). $^1$H NMR (600 MHz, D$_2$O) 6.07 (1H, d, $J = 5.8$ Hz, H1′), 5.33 (1H, d, $J = 5.8$ Hz, H2′), 4.51 (1H, s, H3′), 4.25 (1H, m, H4′), 3.63 (1H, ABX, $J = 12.4$, 5.3 Hz, H5′), 3.51 (1H, ABX, $J = 12.4$, 7.5 Hz, H5″). $^{13}$C NMR (151 MHz, D$_2$O) 190 (C2), 92.5 (C2′), 90.5 (C1′), 87.8 (C4′), 75.2 (C3′), 61.6 (C5′). HRMS (m/z) calculated for C$_6$H$_{10}$NO$_4$S [M + H$^+$]$^+$, 192.0331; found, 192.0332. Crystallographic and refinement parameters are shown in Supplementary Table 3.

**Xylofuranosyl oxazolidinone thione (xylo-1a).** Overall, 46% from xylose (Supplementary Methods) after 14 days at 60 °C (Supplementary Fig. 56). M.p. 133–135 °C (Lit. 129–130 °C[59]). IR (Solid, cm$^{-1}$) 3,251 (OH), 2,930 (CH), 1,488 (C=S). $^1$H NMR (600 MHz, D$_2$O) 6.07 (1H, d, $J = 5.5$ Hz, H1′), 5.29 (1H, d, $J = 5.5$ Hz, H2′), 4.52 (1H, d, $J = 2.8$ Hz, H3′), 4.04 (1H, ddd, $J = 7.4$, 4.3, 2.8 Hz, H4′), 3.96 (1H, ABX, $J = 12$, 4.3 Hz, H5′), 3.84 (1H, ABX, $J = 12$, 7.4 Hz, H5″). $^{13}$C NMR (151 MHz, D$_2$O) 190.3 (C2), 91.1 (C2′), 89.5 (C1′), 80.8 (C4′), 73.5 (C3′), 59.5 (C5′). HRMS (m/z) calculated for C$_6$H$_{10}$NO$_4$S [M + H$^+$]$^+$, 192.0331; found, 192.0332. Crystallographic and refinement parameters are shown in Supplementary Table 4.

**Lyxopyranosyl oxazolidinone thione (p-lyxo-1a).** Overall, 20% from lyxose (Supplementary Methods) after 7 days at 60 °C (Supplementary Fig. 57)[60]. M.p. 185–187 °C (Lit 178 °C[59]). IR (Solid, cm$^{-1}$) 3,342 (NH), 3,237, 3,130 (OH), 2,975 (CH), 1,496 (C=S). $^1$H NMR (600 MHz, D$_2$O) 5.74 (1H, d, $J = 6.1$ Hz, H1′), 5.16 (1H, dd, $J = 6.1$, 3.2 Hz, H2′), 4.02 (1H, dd, $J = 7.5$, 3.2 Hz, H3′), 3.95–3.89 (2H, m, H4′ H5′), 3.65–3.70 (1H, m, H5″). $^{13}$C NMR (151 MHz, D$_2$O) 191.4 (C2), 84 (C2′), 83.8 (C1′), 70.3 (C3′), 68.9 (C4′), 66.2 (C5′). HMBC was observed between H1′ and C5′ indicating pyranosyl structure. HRMS (m/z) calculated for C$_6$H$_{10}$NO$_4$S [M + H$^+$]$^+$, 192.0331; found, 192.0328. Crystallographic and refinement parameters are shown in Supplementary Table 3.

**Lyxofuranosyl oxazolidinone thione (f-lyxo-1a).** $^1$H NMR (600 MHz, D$_2$O) 5.87 (1H, d, $J = 5.8$ Hz, H1′), 5.40 (1H, $t$, $J = 5.8$ Hz, H2′), 4.70 (1H, dd, $J = 6.4$, 5.8 Hz, H3′), 4.31 (1H, ddd, $J = 9$, 6.4, 3.9 Hz, H4′), 3.81 (1H, ABX, $J = 12.4$, 3.9 Hz, H5′), 3.56 (1H, ABX, $J = 12.4$, 9 Hz, H5″). Crystallographic and refinement parameters are shown in Supplementary Table 3.

**(S-Z-cyanovinyl)-arabinofuranosyl-oxazolidinone thione (arabino-1b).** Arabinofuranosyl-oxazolidinone thione (arabino-**1a**; 2.50 g, 13.1 mmol) was added to an aqueous solution of cyanoacetylene (**8**; 20 ml, 1 M). After stirring the reaction mixture for 1 h at room temperature, it was lyophilized to yield (S-Z-cyanovinyl)-arabinofuranosyl-oxazolidinone thione (arabino-**1b**; 3.26 g, quant.) as a white solid, which was used without further purification (Supplementary Fig. 59). An analytical sample was crystallized from ethanol/water (1:4). M.p. 197–203 °C. IR (solid, cm$^{-1}$) 3,372 (OH), 3,068, 3,056, 2,956 (CH), 2,212 (C≡N), 1,601 (C=N), 1,575 (C=C). $^1$H NMR (600 MHz, D$_2$O) 7.83 (1H, d, $J = 10.5$ Hz, H4), 6.18 (1H, d, $J = 6$ Hz, H1′), 5.97 (1H, d, $J = 10.5$ Hz, H5), 5.20 (1H, d, $J = 6$ Hz, H2′), 4.43 (1H, m, H3′), 4.13 (1H, m, H4′), 3.62 (1H, ABX, $J = 12.4$, 4.6 Hz, H5′), 3.53 (1H, ABX, $J = 12.4$, 6.3 Hz, H5″). $^{13}$C NMR (151 MHz, D$_2$O) 167.3 (C2), 143.5 (C5), 115.9 (C6), 100.3 (C4, C1′), 91.5 (C2′), 86.4 (C4′), 76 (C3′), 61.5 (C5′). HRMS (m/z) calculated for C$_9$H$_{10}$N$_2$O$_4$S [M − H$^+$]$^-$, 243.0440; found, 243.0447. Crystallographic and refinement parameters are shown in Supplementary Table 4.

**(S-Z-cyanovinyl)-ribofuranosyl oxazolidinone thione (ribo-1b).** Ribofuranosyl oxazolidinone thione (ribo-**1a**; 331 mg, 1.73 mmol) was added to an aqueous solution of cyanoacetylene (**8**; 5 ml, 1 M). After stirring the reaction mixture for 1 h at room temperature, it was lyophilized to yield (S-Z-cyanovinyl)-ribofuranosyl oxazolidinone thione (ribo-**1b**; 400 mg, 1.65 mmol, 95%) as a white solid, which was used without further purification (Supplementary Fig. 58). M.p. 193–198 °C.

IR (solid, cm$^{-1}$) 3,344 (OH), 3,055, 2,926, 2,876 (CH), 2,218 (C≡N), 1,653 (C=N), 1,601 (C=C). $^{1}$H NMR (600 MHz, D$_2$O) δ 7.88 (1H, d, $J$ = 10.8 Hz, H4), 6.10 (1H, d, $J$ = 5.4 Hz, H1′), 5.99 (1H, d, $J$ = 10.8 Hz, H5), 5.22 (1H, $t$, $J$ = 5.4 Hz, H2′), 4.27 (1H, dd, $J$ = 9.3, 5.4 Hz, H3′), 3.95 (1H, ABX, $J$ = 13, 2.3 Hz, H5′), 3.76 (1H, ABX, $J$ = 13, 4.7 Hz, H5″), 3.60 (1H, ddd, $J$ = 9.3, 4.7, 2.3 Hz, H4′). $^{13}$C NMR (151 MHz, D$_2$O) δ 168.4 (C2), 143.7 (C5), 116 (C6), 100.3 (C4), 98.9 (C1′), 84.7 (C2′), 78.9 (C4′), 70.9 (C3′), 59.9 (C5′). [M − H$^{+}$]$^{−}$ C$_9$H$_{10}$N$_2$O$_4$S calcd 243.0493, found 243.0541.

### Arabinofuranosyl-(2-thiomethyl)-oxazolidinone (arabino-1c).
(S-Z-cyanovinyl)-arabinofuranosyl-oxazolidinone thione (arabino-1b; 24.2 mg, 0.1 mmol) and DSS (NMR standard, 15 mg) were dissolved in D$_2$O (500 μl). Sodium acetate buffer (500 μl, 2 M, pD 6, D$_2$O) was added and the solution adjusted to pD 6 with 1 M NaOD/DCl. Methanethiol (generated by dropping sodium thiomethoxide solution (21% v/v) onto anhydrous monobasic sodium phosphate at room temperature) was bubbled through the solution for 10 min. Further methanethiol gas was added every 2 h. After 6 h the solution was sparged with nitrogen, then cyanoacetylene (8; 200 μl, 1 M) was added and the mixture was left to stand for 2 h. The solution was again saturated with methanethiol gas for 10 min and incubated for 16 h before NMR spectra were acquired (Supplementary Figs 13, 14 and 60). NMR analysis indicated the presence of arabinofuranosyl-(2-thiomethyl)-oxazolidinone (arabino-1c, 50%, Supplementary Methods), arabinofuranosyl-oxazolidinone thione (arabino-1a, 15%) and arabinofuranosyl-oxazolidinone (arabino-22, 12%, Supplementary Methods).

### N3-arabinofuranosyl-2,2′-anhydroquinazolinedione (arabino-12a).
(S-Z-cyanovinyl)-arabinofuranosyl-oxazolidinone thione (arabino-1b; 100 mg, 0.40 mmol) was added to a solution of anthranilic acid (14; 170 mg, 1.20 mmol) in water (10 ml) at pH 6. The reaction was stirred for 18 h at room temperature (Supplementary Fig. 15), after this time the crude was lyophilized and purified by FCC eluting with (EtOAc:MeOH 9:1) to yield N3-arabinofuranosyl-2,2′-anhydroquinazolinedione (arabino-12a; 40 mg, 0.14 mmol, 36%) as a white solid (Supplementary Fig. 64). M.p. 234–238 °C. IR (solid, cm$^{-1}$) 3,422 (OH), 3,067, 2,959, 2,929, 2,875 (CH), 1,699 (C=O), 1,644 (C=N), 1,608 (C–N), 1,562 (C=C). $^{1}$H NMR (600 MHz, D$_2$O) δ 8.15 (1H, d, $J$ = 8.3 Hz, H–Ar), 7.84 (1H, $t$, $J$ = 7.7 Hz, H–Ar), 7.48 (2H, m, H–Ar), 6.74 (1H, d, $J$ = 5.8 Hz, H1′), 5.43 (1H, d, $J$ = 5.8 Hz, H2′), 4.69 (1H, s, H3′), 4.40 (1H, br s, H4′), 3.59 (1H, ABX, $J$ = 13, 3.5 Hz, H5′), 3.57 (1H, ABX, $J$ = 13, 4.3 Hz, H5″). $^{13}$C NMR (151 MHz, D$_2$O) δ 162.7 (C4), 156 (C2), 148.6 (C6), 136.9 (C8), 127.3 (C10), 126.3 (C9), 125.7 (C7), 118.3 (C5), 89.9 (C4′), 89 (C2′), 88.4 (C1′), 75.8 (C3′), 61.5 (C5′). HRMS (m/z): [M$^{+}$] C$_{13}$H$_{12}$N$_2$O$_5$ calcd 276.07462, found 276.07395. Crystallographic and refinement parameters are shown in Supplementary Table 4.

### N3-ribofuranosyl-2,2′-anhydroquinazolinedione (ribo-12a).
(S-Z-cyanovinyl)-ribofuranosyl oxazolidinone thione (ribo-1b; 30.3 mg, 0.125 mmol) was added to a solution of anthranilic acid (14; 34.3 mg, 0.25 mmol) in water (0.5 ml) at pH 6. The reaction was stirred for 18 h at room temperature (Supplementary Fig. 16), then left to stand at 5 °C overnight. The obtained crystals were filtered, washed with water (0.5 ml), ether (5 ml) and dried under vacuum to give N3-ribofuranosyl-2,2′-anhydroquinazolinedione (ribo-12a; 10 mg, 0.036 mmol, 29%) as a white solid. An analytical sample was recrystallized from hot water (Supplementary Fig. 63). M.p. 230–235 °C, decomp. IR (solid, cm$^{-1}$) 3,431 (OH), 3,170 (OH), 2,927 (CH), 1,697 (C=O), 1,645 (C=N), 1,608 (C–N), 1,565 (C=C). $^{1}$H NMR (600 MHz, D$_2$O) δ 8.22 (1H, d, $J$ = 8.1 Hz, H–Ar), 7.91 (1H, $t$, $J$ = 8.1 Hz, H–Ar), 7.58 (1H, d, $J$ = 8.1 Hz, H–Ar), 7.55 (1H, $t$, $J$ = 8.1 Hz, H–Ar), 6.71 (1H, d, $J$ = 5.4 Hz, H1′), 5.52 (1H, $t$, $J$ = 5.4 Hz, H2′), 4.49 (1H, dd, $J$ = 9.4, 5.4 Hz, H3′), 4.02 (1H, ABX, $J$ = 13, 2.3 Hz, H5′), 3.98 (1H, ddd, $J$ = 9.4, 4.1, 2.3 Hz, H4′), 3.85 (1H, ABX, $J$ = 13, 4.1 Hz, H5″). $^{13}$C NMR (151 MHz, D$_2$O) δ 162.9 (C4), 156.5 (C2), 148.8 (C6), 137 (C8), 127.4 (C10), 126.4 (C9), 126 (C7), 118.6 (C5), 87 (C1′), 82 (C2′), 80.6 (C4′), 70.2 (C3′), 59.7 (C5′). HRMS (m/z): [M + H$^{+}$]$^{+}$ C$_{13}$H$_{12}$N$_2$O$_5$ calcd 277.0825, found 277.0820. Crystallographic and refinement parameters are shown in Supplementary Table 5.

### Arabinofuranosyl-N-acetonitrile-aminooxazoline (arabino-10d).
Glycine nitrile.HCl (2d.HCl; 2.7 g, 29.2 mmol) was dissolved in water (10 ml) and the solution was adjusted to pH 4.5 with NaOH (4 M) (Supplementary Fig. 8). Arabinofuranosyl-(2-thiomethyl)-oxazolidinone (arabino-1c; 2 g, 9.78 mmol) was added in a single portion and the solution was re-adjusted to pH 4.5. The reaction was stirred for 2 h at room temperature whilst N$_2$ constantly agitated the solution (Supplementary Figs 20 and 21). The solution was neutralized with sodium hydroxide (4 M), and silica gel (5 g) was added. The suspension was concentrated to a free flowing powder that was purified by flash column chromatography (FCC), eluting with MeOH/CHCl$_3$ (1:9–2:8). The fractions containing the product were concentrated to yield arabinofuranosyl-N-acetonitrile-aminooxazoline (arabino-10d; 1.27 g, 5.96 mmol, 61%) as a yellow oil (Supplementary Fig. 62). IR (neat, cm$^{-1}$) 3,246 (OH), 2,926 (CH), 2,251 (CN), 1,648 (C=N). $^{1}$H NMR (600 MHz, D$_2$O) 6 (1H, d, $J$ = 5.5 Hz, H1′), 5.04 (1H, d, $J$ = 5.5 Hz, H2′), 4.38 (1H, d, $J$ = 2.8 Hz, H3′), 4.21 (2H, s, CH2), 4.07 (1H, ddd, $J$ = 6.6, 5.5, 2.8 Hz, H4′), 3.60 (1H, ABX, $J$ = 12.2, 5.5 Hz, H5′),

3.55 (1H, ABX, $J$ = 12.2, 6.6, H5″). $^{13}$C NMR (151 MHz, D$_2$O) 163.3 (C2), 118.3 (C5), 97.6 (C1′), 89.8 (C2′), 85.9 (C4′), 75.8 (C3′), 61.9 (C5′), 31.8 (C4). HRMS (m/z) calculated for C$_8$H$_{12}$N$_3$O$_4$ [M + H$^{+}$]$^{+}$, 214.0828; found, 214.0813.

### 2,2′-Anhydro-5-aminoimidazole-4-carboxamide-β-furanosylarabinoside (16c).
To a solution of 2-aminocyanoacetamide (2c; 108 mg, 1.09 mmol) in water (1 ml) at pH 4.5 was added arabinofuranosyl-(2-thiomethyl)-oxazolidinone (arabino-1c; 300 mg, 1.46 mmol). The resultant mixture was stirred for 3 h at 45 °C and the pH of the reaction was kept between 4.3 and 4.7 during the course of the reaction. After 3 h the pH of the reaction was raised to pH 8 and then left to stir overnight at 45 °C (Supplementary Figs 22–26). After 24 h white solids precipitated from the crude mixture, which were filtered, washed and dried to yield 2,2′-anhydro-5-aminoimidazole-4-carboxamide-β-furanosylarabinoside (16c; 37.2 mg, 0.15 mmol, 13%) as white solid (Supplementary Fig. 67). M.p. 234–238 °C. IR (solid, cm$^{-1}$) 3,493 (H$_2$N–C=O), 3,359 (H$_2$N–C), 3,210 (OH), 1,638 (C=O), 1,581 (C=N), 1,533 (C=C). $^{1}$H NMR (600 MHz, D$_2$O) 6.46 (1H, d, $J$ = 5.4 Hz, H1′), 5.68 (1H, d, $J$ = 5.4 Hz, H2′)), 4.60 (1H, br s, H3′), 4.38 (1H, ddd, $J$ = 6.8, 4.9, 1.9 Hz, H4′), 3.56 (1H, ABX, $J$ = 12.4, 4.9 Hz, H5′), 3.42 (1H, ABX, $J$ = 12.4, 6.8 Hz, H5″). $^{13}$C NMR (151 MHz, D$_2$O) 169.1 (C6), 152.6 (C2), 140 (C5), 109.9 (C4), 97.9 (C2′), 89.3 (C4′), 86.5 (C1′), 75.3 (C3′), 61.5 (C5′). HRMS (m/z) calculated for C$_9$H$_{12}$N$_4$O$_5$ [M + H$^{+}$]$^{+}$, 257.0880; found, 257.0882. Crystallographic and refinement parameters are shown in Supplementary Tables 5 (D-isomer) and 6 (L-isomer).

### 2,2′-Anhydro-5-aminoimidazole-4-carbonitrile-β-furanosylarabinoside (16b).
Aminomalonitrile p-toluenesulfonate (126.7 mg, 0.5 mmol) was dissolved in D$_2$O (450 μl) and the pD was adjusted to 3 with NaOD (4 M). A solution of DSS (D$_2$O, 50 μl, 244 mM) was added followed by arabinofuranosyl-(2-thiomethyl)-oxazolidinone (arabino-1c; 25.6 mg, 0.125 mmol). The pD was re-adjusted to 3 and the mixture was stirred at room temperature for 3 h. An aliquot (50 μl) was added to ammonium hydroxide in D$_2$O (450 μl, 100 mM, pD 9) and the solution was adjusted to pD 9 with NaOD (4 M) and incubated for 1 h at room temperature (Supplementary Figs 27 and 28). Spiking with authentic samples and calibration to an internal standard confirmed yield of 15% 2,2′-anhydro-5-aminoimidazole-4-carbonitrile-β-furanosylarabinoside (16b; Supplementary Methods). M.p. 210–220 °C decomp. IR (Solid, cm$^{-1}$) 3,423 (H$_2$N–C), 3,316 (OH), 2,194 (CN), 1,649 (C=N), 1,568 (C=C). $^{1}$H NMR (600 MHz, DMSO) 6.55 (1H, br s, NH$_2$), 6.19 (1H, d, $J$ = 5.2 Hz, H1′), 5.82 (1H, br s, 3′OH), 5.46 (1H, d, $J$ = 5.2 Hz, H2′), 4.98 (1H, $t$, $J$ = 5.5 Hz, 5′OH), 4.31 (1H, br s, H3′), 3.98 (1H, ddd, $J$ = 7.3, 5.5, 2.2 Hz, H4′), 3.22 (1H, ABXY, $J$ = 11.2, 5.5, 5.5 Hz, H5′), 3.11 (1H, ABXY, $J$ = 11.2, 7.3, 5.3 Hz, H5′). $^{13}$C NMR (151 MHz, DMSO) 150 (C2), 143.8 (C4), 117.8 (C6), 97.4 (C2′), 88.4 (C4′), 86.6 (C4), 85.3 (C1′), 74 (C3′), 60.6 (C5′). HRMS (m/z) calculated for C$_9$H$_{10}$N$_4$O$_4$ [M + H$^{+}$]$^{+}$; 239.07748; found, 239.07750. Crystallographic and refinement parameters are shown in Supplementary Table 5 and Supplementary Fig. 66).

### 8,2′-O-anhydro-9-β-arabinofuranosyl-cycloadenosine (17A).
Method A. 2,2′-Anhydro-5-aminoimidazole-4-carbonitrile-β-furanosylarabinoside (16b; 4.8 mg, 0.02 mmol) and DSS (internal standard, 5 mg) were dissolved in D$_2$-formamide (500 μl). The mixture was heated at 100 °C and NMR spectra were periodically acquired. The starting material had been consumed after 96 h. The reaction mixture was cooled to room temperature, lyophilized and the residue dissolved in methanolic ammonia (to remove 3′ and 5′ formyl groups observed during reaction in neat formamide) for 30 min. The reaction was concentrated to dryness, D$_2$O (5 ml) was added and the mixture was lyophilized. The residue was the dissolved in $d_6$-DMSO and NMR spectra were acquired. Spiking with an authentic sample of 8,2′-O-anhydro-9-β-arabinofuranosyl-cycloadenosine (17A, Supplementary Methods) and calibration to the internal standard (DSS) confirmed a 10% yield of 17A.

Method B. 2,2′-Anhydro-5-aminoimidazole-4-carbonitrile-β-furanosylarabinoside (16b; 4.8 mg, 0.02 mmol), formamidine hydrochloride (16.1 mg, 0.2 mmol) and DSS (internal standard, 5 mg) were dissolved in D$_2$-formamide (500 μl). The mixture was heated at 100 °C and NMR spectra were periodically acquired. The starting material had been consumed after 5 h. The reaction mixture was cooled to room temperature and lyophilized. Formamidine was observed to suppress 3′ and 5′ formylation and no methanolic ammonia treatment was required. The lyophilite was dissolved in D$_2$O (5 ml) and again lyophilized. The residue was the dissolved in $d_6$-DMSO and NMR spectra were acquired (Supplementary Figs 29–32). Spiking with an authentic sample of 8,2′-O-anhydro-9-β-arabinofuranosyl-cycloadenosine (17A, Supplementary Methods) and calibration to the internal standard (DSS) confirmed a 65% yield of 17A (Supplementary Fig. 68). M.p. 205 °C decomp. (Lit.[61] 210 °C) IR (Solid, cm$^{-1}$) 3,297 (NH$_2$), 3,142 (OH), 2,962, 2,883 (CH), 1,667 (C=C), 1,623 (C=N). $^{1}$H NMR (600 MHz, D$_2$O) 8.16 (1H, s, H2), 6.71 (1H, d, $J$ = 5.5 Hz, H1′), 5.90 (1H, d, $J$ = 5.5 Hz, H2′), 4.72 (1H, s, H3′), 4.40 (1H, ddd, $J$ = 5.5, 5.4, 4.3 Hz, H4′), 3.55 (1H, ABX, $J$ = 12.7, 4.3 Hz, H5′), 3.48 (1H, ABX, $J$ = 12.7, 5.4 Hz, H5′). $^{13}$C NMR (151 MHz, D$_2$O) δ 161.2 (C8), 154.4 (C5), 151.6 (C2), 146.2 (C4), 121.1 (C6), 99.5 (C2′), 89.6 (C4′), 86.5 (C1′), 75.6 (C3′), 61.4 (C5′). HRMS (m/z): [M − H$^{+}$]$^{−}$ C$_{10}$H$_{11}$N$_5$O$_4$ calcd 265.08110, found 266.0866. Crystallographic and refinement parameters are shown in Supplementary Table 6.

**8,2′-O-anhydro-9-β-arabinofuranosyl-cycloinosine (17I).** 2,2′-Anhydro-5-aminoimidazole-4-carboxamide-β-furanosylarabinoside (**16c**; 5 mg, 0.02 mmol) and DSS (NMR standard, 5 mg) were dissolved in $D_2$-formamide (500 µl). The mixture was heated at 100 °C and NMR spectra were periodically acquired. The starting material had been consumed after 72 h. The reaction mixture was cooled to room temperature, lyophilized and the residue was treated with methanolic ammonia (to remove 3′ and 5′ formyl groups) for 30 min. The reaction was concentrated to dryness, $D_2O$ (5 ml) was added and the mixture was lyophilized. The residue was dissolved in $D_6$-DMSO and NMR spectra were acquired (Supplementary Fig. 33). Spiking with an authentic sample of 8,2′-O-anhydro-9-β-arabinofuranosyl-cycloinosine (**17I**, Supplementary Methods) and calibration to the internal standard (DSS) confirmed a 3% yield of **17I** (Supplementary Fig. 69). The addition of formamidine hydrochloride (16.1 mg, 0.2 mmol) at the start of the reaction increases the yield of **17I** to 11%. M.p. 255 °C (decomp). IR (Solid, cm$^{-1}$) 3,216 (OH) or (NH), 3,076, 2,006, 2,953 (CH), 1,673 (C=O), 1,598 (C=C), 1,572 (C=N). $^1$H NMR (600 MHz, $D_2O$) δ 8.17 (1H, s, H2), 6.75 (1H, d, J = 5.5 Hz, H1′), 5.91 (1H, d, J = 5.5 Hz, H2′), 4.72 (1H, s, H3′), 4.42 (1H, dd, J = 5.4, 4.1 Hz, H4′), 3.56 (1H, ABX, J = 12.7, 4.1 Hz, H5′), 3.49 (1H, ABX, J = 12.7, 5.4 Hz, H5′). $^{13}$C NMR (151 MHz, $D_2O$) δ 160.7 (C8), 158.4 (C6), 145.8 (C2), 145.6 (C5), 126 (C4), 99.5 (C2′), 89.9 (C4′), 86.9 (C1′), 75.6 (C3′), 61.4 (C5′). HRMS (m/z): [M-H$^+$]$^-$ $C_{10}H_{10}N_5O_4$ calcd 267.0729, found 267.0723. Crystallographic and refinement parameters are shown in Supplementary Table 6.

**Phosphorylation of anhydronucleosides.** *Method I.* Nucleoside (0.06 mmol), ammonium dihydrogen phosphate (0.06 mmol) and urea (1.6 mmol) were thoroughly mixed and heat at 140 °C for 20 min (Supplementary Table 2). The reaction mixture was then cooled to room temperature, dissolved in $D_2O$ (3 ml) and lyophilized. The residue was thrice dissolved in $D_2O$ (3 ml) and lyophilized. The final lyophilite was dissolved in $D_2O$ (0.5 ml) and NMR spectra were acquired (Supplementary Figs 35 and 45). Yields were calculated based on comparison with an internal standard (DSS) and nucleotide phosphates were purified using HPLC (Supplementary Methods).

*Method II.* Nucleoside (0.06 mmol), ammonium dihydrogen phosphate (0.06 mmol) and urea (0.6 mmol) were suspended in formamide (0.6 ml). The reaction mixture was heated at 100 °C for 72 h. The reaction was diluted $D_2O$ (3 ml) and lyophilized for 3 days (to remove formamide), diluted with $D_2O$ (3 ml) and further lyophilized. The lyophilite was dissolved in $D_2O$ (0.5 ml) and NMR spectra were acquired (Supplementary Figs 42 and 52). Yields were calculated based on comparison with an internal standard (DSS).

*Method III.* Nucleoside (0.06 mmol), ammonium dihydrogen phosphate (0.06 mmol) and urea (1.6 mmol) were thoroughly mixed and heat at 140 °C for 20 min. Glycerol (44 µl, 0.6 mmol) was added and the mixture was heated at 140 °C for a further 20 min. The reaction mixture was then cooled to room temperature, dissolved in $D_2O$ (3 ml) and lyophilized. The residue was thrice dissolved in $D_2O$ (3 ml) and lyophilized. The final lyophilite was dissolved in $D_2O$ (0.5 ml) and NMR spectra were acquired (Supplementary Fig. 43). Yields were calculated based on comparison with an internal standard (DSS).

*Method IV.* Nucleoside (0.06 mmol), ammonium dihydrogen phosphate (0.06 mmol) and urea (1.6 mmol) were thoroughly mixed and heat at 140 °C for 20 min. Cytidine (14.6 mg, 0.06 mmol) was added and the mixture was heated at 140 °C for a further 20 min. The reaction mixture was then cooled to room temperature, dissolved in $D_2O$ (3 ml) and lyophilized. The residue was thrice dissolved in $D_2O$ (3 ml) and lyophilized. The final lyophilite was dissolved in $D_2O$ (0.5 ml) and NMR spectra were acquired (Supplementary Fig. 43). Yields were calculated based on comparison with an internal standard (DSS).

*Method V.* Nucleoside (0.03 mmol), ancitabine (**11**; 0.03 mmol), ammonium dihydrogen phosphate (0.06 mmol) and urea (0.6 mmol) were suspended in formamide (0.6 ml). The reaction mixture was heated at 100 °C for 72 h. The reaction was diluted $D_2O$ (3 ml) and lyophilized for 3 days (to remove formamide), diluted with $D_2O$ (3 ml) and further lyophilized. The lyophilite was dissolved in $D_2O$ (0.5 ml) and NMR spectra were acquired (Supplementary Fig. 44). Yields were calculated based on comparison with an internal standard (DSS).

**8-Oxo-adenosine-2′,3′-cyclic phosphate (3OA).** M.p. 220 °C (decomp.). IR (Solid, cm$^{-1}$) 1,711 (N–C=N), 1,649 (N–C=O), 1,060 (P=O). $^1$H NMR (600 MHz, $D_2O$) 8.10 (1H, s, H2), 6.11 (1H, d, J = 3.4 Hz, H1′), 5.61 (1H, ddd, J = 7.8, 7, 3.4 Hz, H2′), 5.14 (1H, ddd, J = 10.9, 7, 4.9 Hz, H3′), 4.40 (1H, m, H4′), 3.93 (1H, ABX, J = 12.3, 3.6 Hz, H5′), 3.85 (1H, ABX, J = 12.3, 5.5 Hz, H5″). $^{31}$P NMR (162 MHz, $D_2O$) 20.37 (dd, J = 10.9, 7.8 Hz, 2′,3′cP). $^{13}$C NMR (151 MHz, $D_2O$) 153.3 (C8), 151.8 (C2), 148.3 (C6), 146.7 (C4), 105.2 (C5), 87 (d, J = 5.5 Hz, C1′), 85.6 (d, J = 2.2 Hz, C4′), 79.8 (d, J = 2.2 Hz, C2′), 78.3 (C3′), 62 (C5′). HRMS (m/z): [M – H$^+$]$^-$ for $C_{10}H_{12}N_5O_7P$ calcd 344.0396, found 344.0390. (Supplementary Figs 36–38).

**8-Oxo-adenosine-2′,3′-cyclic-5′ bisphosphate (18OA).** M.p. 235 °C (decomp.). IR (Solid, cm$^{-1}$) 1,707 (N–C=N), 1,655 (N–C=O), 1,038 (P=O). $^1$H NMR (600 MHz, $D_2O$) 8.12 (1H, s, H2), 6.11 (1H, d, J = 2.9 Hz, H1′), 5.69 (1H, td, J = 6.3, 2.9 Hz, H2′), 5.19 (1H, ddd, J = 12, 6.3, 5.8 Hz, H3′), 4.44 (1H, m, H4′), 4.14 (1H, ABXY, J = 11.4, 5, 5 Hz, H5′), 4.06 (1H, ABXY,

$J = 11.4, 6.2, 6.2$ Hz, H5″). $^{31}$P NMR (162 MHz, $D_2O$) 20.58 (dd, J = 12, 6.3 Hz, 2′,3′cP), 0.85 (br s, 5′P). $^{13}$C NMR (151 MHz, $D_2O$) 153.2 (C8), 151.9 (C2), 148.2 (C6), 146.8 (C4), 105.1 (C5), 86.7 (d, J = 6.6 Hz, C1′), 84.4 (d, J = 7.2 Hz, C4′), 79.4 (d, J = 2.8 Hz, C2′), 77.9 (C3′), 64.5 (d, J = 3.9 Hz, C5′). HRMS (m/z): [M – H$^+$]$^-$ for $C_{10}H_{13}N_5O_{10}P_2$ calcd 424.0060, found 424.0056. (Supplementary Figs 39–41).

**8-Oxo-inosine-2′,3′-cyclic phosphate (3OI).** M.p. 310 °C (decomp.). IR (Solid, cm$^{-1}$) 1,724 (C=O), 1,674 (N–C=O), 1,447 (C=C), 1,063 (P=O). $^1$H NMR (600 MHz, $D_2O$) 8.14 (1H, s, H2), 6.16 (1H, d, J = 2.9 Hz, H1′), 5.62 (1H, td, J = 6.6, 2.9 Hz, H2′), 5.17 (1H, ddd, J = 5.4, 6.6, 12.1 Hz, H3′), 4.38 (1H, m, H4′), 3.92 (1H, ABX, J = 12.4, 3.9 Hz, H5′), 3.85 (1H, ABX, J = 12.4, 5.9 Hz, H5″). $^{31}$P NMR (162 MHz, $D_2O$) 20.44 (dd, J = 12.1, 6.6 Hz, 2′,3′cP). $^{13}$C NMR (151 MHz, $D_2O$) 153.3 (C6), 153.2 (C8), 145.9 (C2), 145.7 (C4), 109.6 (C5), 87 (d, J = 6.6 Hz, C1′), 85.7 (d, J = 1.7 Hz, C4′), 80.2 (d, J = 2.2 Hz, C2′), 78.2 (C3′), 61.8 (C5′). HRMS (m/z): [M – H$^+$]$^-$ for $C_{10}H_{11}N_4O_8P$ calcd 345.0236, found 345.0235. (Supplementary Figs 46–48).

**8-Oxo-inosine-2′,3′-cyclic-5′ bisphosphate (18OI).** M.p. 275 °C (decomp.). IR (Solid, cm$^{-1}$) 1,680 (N–C=O), 1,452 (C=C), 1,059 (P=O). $^1$H NMR (600 MHz, $D_2O$) 8.15 (1H, s, H2), 6.16 (1H, d, J = 2.8 Hz, H1′), 5.67 (1H, td, J = 6.4, 2.8 Hz, H2′), 5.22 (1H, ddd, J = 12.6, 6.4, 5.7 Hz, H3′), 4.45 (1H, m, H4′), 4.10 (1H, ABXY, J = 11.3, 4.9, 4.9 Hz, H5′), 4.03 (1H, ABXY, J = 11.3, 6.1, 6.1 Hz, H5″). $^{31}$P NMR (162 MHz, $D_2O$) 20.59 (dd, J = 12.6, 6.4 Hz, 2′,3′cP), 2.77 (br s, 5′P). $^{13}$C NMR (151 MHz, $D_2O$) 153.5 (C6), 153.2 (C8), 146.1 (C2), 145.8 (C4), 109.8 (C5), 86.8 (d, J = 6.6 Hz, C1′), 84.7 (d, J = 8.3 Hz, C4′), 79.9 (d, J = 2.8 Hz, C2′), 78.2 (C3′), 64.4 (d, J = 4.4 Hz, C5′). HRMS (m/z): [M-H$^+$]$^-$ for $C_{10}H_{12}N_4O_{11}P_2$ calcd 424.9900, found 424.9904. (Supplementary Figs 49–51).

**Crystallography.** Single X-ray diffraction data for **4b**, arabino-**1a**, lyxo-furano-**1a**, lyxo-pyrano-**1a**, ribo-**1a**, arabino-**1b**, arabino-**12a**, ribo-**12a**, xylo-**1a** and **15** were collected using a Bruker APEX II DUO CCD diffractometer (CuK$_\alpha$ radiation, $\lambda = 1.54178$ Å or MoK$_\alpha$, $\lambda = 0.71073$ Å). The data were processed using the Bruker SAINT V7.46A software[62]. All data collections were performed at low temperatures. Diffraction data for **16b**, L-**16c**, D-**16c**, **17A**, **17I**, **19** and **20** were collected using an Agilent SuperNova (Dual Source) single crystal X-ray diffractometer (CuK$_\alpha$ radiation, $\lambda = 1.54184$ Å). The data were processed using the CrysAlisPro programme[63]. All diffraction data was collected at low temperatures. The crystal structures were solved by the direct methods procedure (using either SHELXS-97 or SHELXS-2014/7)[64] and refined by least-squares methods against F2 (using SHELXS-97 or SHELXL-2014/7)[64]. All non-hydrogen atoms were refined anisotropically. Hydrogen atoms affiliated with oxygen and nitrogen atoms were refined isotropically in positions identified in the difference Fourier map or in geometrically constrained positions, while hydrogen atoms associated with carbon atoms were refined isotropically in geometrically constrained positions. Where possible the absolute configuration of each structure was determined through anomalous dispersion effects observed in diffraction measurements of chiral compound (Supplementary Crystallographic Files), for any results that were not conclusive, owing to the lack of heavy atoms in the molecule, the absolute configuration was established by reference to an unchanged chiral center in the synthetic sequence. The molecular structures of **4b**, lyxo-furano-**1a**, lyxo-pyrano-**1a**, ribo-**1a**, xylo-**1a** and L-arabino-**16c** are shown in Supplementary Fig. 53, while the molecular structures of arabino-**12a**, ribo-**12a** are shown in Fig. 3, the molecular structures of **15**, arabino-**1a**, arabino-**1b**, arabino-**16b**, D-arabino-**16c** are shown in Fig. 4, and the molecular structures of **17A**, **17I**, **19** and **20** are shown in Fig. 6. Essential crystallographic and refinement parameters relating to the reported crystal structures are summarized in Supplementary Tables 3–7.

**Data availability.** The authors declare that data supporting the findings of this study are available within the paper and its Supplementary Information files. All relevant data is available from the authors upon reasonable request. X-ray crystallographic data was also deposited at the Cambridge Crystallographic Data Centre (CCDC) under the following CCDC deposition numbers: D-arabino-**1a** (1522010), D-lyxo-furano-**1a** (1522011), D-lyxo-pyrano-**1a** (1522012), D-ribo-**1a** (1522013), D-xylo-**1a** (1522014), D-arabino-**1b** (1522015), **4b** (1522016), D-arabino-**12a** (1522017), D-ribo-**12a** (1522018), **15** (1522019), D-**16b** (1522026), L-**16c** (1522024), D-**17A** (1522025), D-**17I** (1522023), D-**19** (1522021) and D-**20** (1522022). These can be obtained free of charge from CCDC via www.ccdc.cam.ac.uk/data_request/cif.

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

## Acknowledgements

This work was supported in part by the Simons Foundation (318881 to M.W.P. and 290363 to J.W.S.), the Engineering and Physical Sciences Research Council (EP/K004980/1 to M.W.P.) and through an award from the Origin of Life Challenge (to M.W.P.) and a UCL Excellence Fellowship (to D.-K.B.). We thank Dr K. Karu for assistance with Mass Spectrometry and Dr A.E. Aliev for assistance with NMR spectroscopy. J.W.S. is an Investigator of the Howard Hughes Medical Institute.

## Author contributions

M.W.P. conceived the research. M.W.P., J.W.S., S.S. and A.N. designed and analysed the experiments. M.W.P, S.S. and A.N. conducted the experiments. D.-K.B. and S.-L.Z. performed the crystallographic analyses. M.W.P, J.W.S. and S.S. wrote the paper.

## Additional information

**Competing interests:** The authors declare no competing financial interests.

