## [Peer review file · Nature Communications]

Reviewers' comments:

Reviewer #1 (Remarks to the Author):

The paper makes the very interesting and creative point that purines might have been initially different to allow for a common synthesis. This is convincing argument and the manuscript goes in length to show the idea in experiments. It points out that 8-oxo-ribonucleotides could be a common starting point to have pyrimidines and 8-oxo-purines to emerge in the first genetic molecules.

The starting point of the authors is that the incompatibility of synthesizing both purines and pyrimidines stems from the the fact that both display significant variation in their oxidation level. In addition the (albeit perhaps limited) evidence that oxo-purines can pair with the complementary bases also points to the possibility of an intermediate.

This makes a lot of sense also since having sequences with only GC-pair in RNA poses major problems which require a more complex 4-letter code to compensate. So some less strong binding base alternatives for at least for the G purine is something that makes a lot of sense from an evolutionary standpoint. Only later, when the four letter code has balanced itself out, evolution could be ready for the modern GC-pair.

The whole manuscript describes a longer chemical argument, which sometimes looks quite special. On the other hand, a complex route is to be expected for purine synthesis where the simple routes have been already shown not to be successful - and the proposed pathway, at least for the high concentrations taken, is rewarding in terms of yields. The network of synthesis is explored in great detail and does not leave much questions. This is a solid piece of work and the authors in their account give a complete and mostly unbiased account of their arguments and experiments. This style is very pleasing, compact, but sometimes not optimal to find a good starting point. But the authors describe their overall scheme very well.

I like the manuscript and I am much in favor of publishing. Especially the supplement is very carefully made and shows all the details of the reactions in a very transparent way. Still I think the manuscript could be improved by considering the points below for in a revised version of the manuscript:

- The letter starts quite bold with that "Understanding prebiotic nucleotide synthesis is one of the most important challenges in elucidating the origins of life on Earth.". Well, honestly, I am not a big fan that papers have to state that its own mono-disciplinary viewpoint is the most important one - especially in this field where many equally important challenges have to be solved to reconstruct the origin of life. I think this could be tuned down.

- In terms of presentations, the main message of the manuscript could be better presented in Figure 1. I understand that the color coding is used to trace the history of parts, but the main result of co-synthesis of purines and pyrimidines is for non-specialists quite hidden. For example, some more text in Figure 1 and a longer Figure caption could help in guiding the reader.

- That said also the conclusion text is not really well linked and represented in the Figures.

The authors should perhaps appreciate that it is not so easy for readers to get even the broad idea of the manuscript. The manuscript is full of important and interesting details and the detailed main text is well guided by the Figures, but some help in getting the broad picture would be helpful. For example, both the abstract and the introductory text misses links to Figures and structures. So perhaps it would be good to add a Figure 0 with the main message of the paper and link this to abstract and introduction. Right now the text first links to structure (1) from Figure 2 before going in details with Figure 1. Perhaps it is useful to visually highlight the divergence in Figure 1 so that also (1) is found faster on Figure 1. It would be just a pity if the nice results are only appreciated fast by real specialists - even more so since the topic of OoL is very interdisciplinary. Such a change will tremendously increase the impact of the paper.

- While biological results on DNA are used to motivate the intermediate purines and many references link to DNA, there is no discussion on how this scheme would be compatible with a co-synthesis of both RNA and DNA, something that would be not futile even in an RNA world scenario. Perhaps the authors want to comment on this?

Reviewer #2 (Remarks to the Author):

This work should be of high interest to chemists and biologists who follow experimental results in prebiotic chemistry in general, and the abiotic formation of nucleosides, nucleotides and ultimately nucleic acids in particular, which is a very hot topic at the front of opinion making in the origin of life field.

The author's major claims are that, both, pyrimidine nucleosides and nucleotides and purine nucleosides and nucleotides can be formed concomitantly without excluding one another, under plausibly prebiotic conditions, i.e. solubilized in ambient water at pH values between 2 and 11 (usually 6, some reactions up to pH 13) or hot formamide. Many of the reactions are based on known literature data that almost all have been correctly cited, such as the use of thiocyanic acid (**9**), 2-aminomalonoitrile (**2b**), 2-amino-2-cyanoacetamide (**2c**), carbohydrate-derived 1,3-oxazolidine-2-thiones (**1a**, **1c** and **1d**) and the nucleophilic attack by anthranilic acid (**14**). The key steps, however, result from the author's stubbornly systematic testing of slightly modified aqueous reaction conditions that presumably have been actually discovered by the chemists among the authors. Examples: the slightly acidic conditions for an efficient nucleophilic attack of ammonia on thiones rather than basic conditions on S-methyl or S-benzyl thiones, the ring closure of the above thiones using 2-aminomalonoitrile or 2-amino-2-cyanoacetamide but not glycine nitrile to give imidazole anhydronucleosides, or the ring closure on the latter using formamidine to give 2'-anhydro-8-oxypurine nucleosides. The impact of this work will be very strong.

The techniques used by Powner & colleagues are based on the ground-breaking works of John Sutherland (and Matthew Powner), combined with a meticulous and, I daresay, fresh and fearless combination of analyses of complex crude reaction mixtures by nuclear magnetic resonance spectroscopy and X Ray crystallography. The spectroscopic data are solid and supported, almost wherever possible, through a "spiking" of the analytes with synthetic reference compounds, which, to the least, minimizes, or even prevents false or subjective readouts from highly complex mixtures. The supplemental files are very clearly graphically exposed, easy to read and to check, why the authors argued the way they did in the main paper. The main text is well written, although this reviewer feels that in the introduction and conclusion some rephrasing is mandatory for the sake of minimizing subjectiveness and maximizing fairness towards adversaries. Let me explain.

Since almost a year it is known that certain purine nucleosides (inosine and guanosine better than adenosine) can be synthesized under aqueous non-demanding conditions from pyrimidine derivatives in the presence of carbohydrates (Becker et al. cited as ref. 7). These authors thus proposed an alternative pathway to purine nucleosides being however not quite prebiotically convincing, owing to the lack of prebiotic conditions for the synthesis of 4,5,6-triamino-2-thiopyrimidine from thiourea and 2-aminomalonoitrile (**2b**) and the lack of a prebiotically feasible reduction step to give adenosine from 2-thioadenosine. Their work is footed on old German chemistry (Wilhelm Traube 1904) combined with studies directed by Albert Eschenmoser (Trinks 1987 and Koch 1992). Becker et al. explicitly emphasized that their pathway delivered regioselective N-glycosidation, although this absolute regioselectivity had already been shown by Traube and repeated later by followers before Becker et al. 2016.

Still, it does not seem fair to write in the introduction: "A recently proposed pathway to the purine nucleotides is similarly problematic due to low yielding steps that require prebiotically implausible reaction conditions and generate a wide spectrum of isomers." After checking the yields that were reported for the steps in ref. 7, the following values can be written down. PhD Thesis Koch 1992: **2b** to **2c** using formaldehyde at pH 8 = 42 % (isolated 20 %). **2c** + guanidine to 2,5,6-triamino-4-oxopyrimidine hydrosulfate at 40 °C in 3 hours = 53 %. Traube: **2b** + thiourea to 4,5,6-triamino-2-thiopyrimidine in ethanol/sodium ethanolate = 84 % (according to Becker et al. 2016). 2,5,6-triamino-4-oxopyrimidine + formamide to FaPyG = 85 %. 4,5,6-triamino-2-thiopyrimidine + formamide to FaPyA 50 % (and 73 %). No particularly low yields up to here, at least not when compared to the yields in this work. Then: FaPyA + ribose to α P + β P + α fA + β fA = 5-20 %. Here partly low yields can be found. In addition, the "wide spectrum of isomers" are carbohydrate isomers, especially when ribose was replaced by formose-type reactants (glyceraldehyde, glycol aldehyde, formaldehyde) and not N-glycoside isomers, as one might think after a first uncritical reading (suggesting non-regiospecific base-sugar attachment). High total N-glycosidation yields on diastereoisomeric mixtures of tetroses and pentoses on formose-type mixtures containing pyrimidine derivatives should not be seen as a really problematic result. In other words, the above sentence is so compact that it suggests to the reader an outcome of ref. 7 worse than it really was. Please be more precise and thus more fair: Which steps were low yielding, which ones were implausibly prebiotic and the spectrum of which isomeric compounds was wide?

Two sentences later (and, more briefly, in the abstract) it says: "Remarkably, all proposed prebiotic syntheses of pyrimidine and purine nucleotides are mutually incompatible;⁴⁻¹⁰". On what scientific, not intuitive, grounds can the authors conclude that all these syntheses are mutually incompatible? Nobody tested experimentally whether they are or are not compatible. I do not see on theoretical-chemical grounds

any incompatibility, the more so, since some of the used compounds are the same in several studies. Especially **2b** and **2c** were never before tested for triaminopyrimidine and oxazolidine formation in one pot. What would happen if cyanamide (**6**) and/or thiocyanic acid (**9**) were mixed with glycolaldehyde (**5**), 2-aminomalononitrile (**2b**) and/or 2-amino-2-cyanoacetamide (**2c**)? Nobody knows yet, or am I wrong? Maybe I am missing something but, if so, then please explain more precisely this incompatibility. Or else, re-phrased please (in the abstract and the main text). There is no need to justify the own work by some virtual, unproven lack of the works of others. In other words, even if other propositions are mutually compatible, there still would be a scientific need to show that your proposition is mutually compatible as well.

Neither on page 5, in the schemes or in the supplemental file have I found the information on whether racemic or enantiopure glyceraldehyde (**7**) was used and, if enantiopure, which, D or L?

On page 6 it says: "Here, we envisaged cyanoacetylene (**8**) as ideally suited to activate thione **1a** in aqueous solution due the slow reaction of cyanoacetylene (**8**) with water and the excellent electron withdrawing properties of the cyanovinyl moiety.^{28,29}" (-> moiety) At this stage it is not clear why excellent electron withdrawing properties would be important, perhaps for preventing hydrolysis of the anhydro linkage or for a stereochemical inversion?

4 lines later: "Due to the higher pK_a of S-cyanovinyl thione **1b** and sulfur-prohibited annulation, no increase... " Higher pK_a than pK_a of what?

Last sentence of page 6 and first of page 7: "... to our knowledge, thiolate displacement from **1c** (R = Me) by ammonia (**2a**) to yield **10a** has only been observed upon reflux in "undistilled DMF" (which was likely contaminated with ammonia (**2a**) and formic acid).³¹" Careful, dimethylformamide does not produce ammonia and formic acid. In ref. 31 the reactants were heated for 3 hours at 90 °C (not reflux) in "technical grade formamide" (not DMF), which was likely contaminated with ammonium formate that can slowly release ammonia and formic acid. I suggest to correct this and perhaps also cite Pietrucci, F.; Saitta, A. M. *Proc. Natl. Acad. Sci. USA* **2015**, *112*, 15030-15035.

Next sentences: "However, the S-benzyl thione **1d** (R = CH₂Ph) has been substituted during quinazolinone **12** synthesis,³² but reports are limited to anthranilic acid derivatives in organic solvents." Ref. 32 is absolutely key to this work. However, Tatibouët & colleagues have published 1,3-oxazolidine-2-thiones of various carbohydrates three years before already. Please cite (in addition to or replace with) Girimene, J.; Gueyraud, D.; Tatibouët, A.; Rollin, P. *Tetrahedron* **2001**, *42*, 2977-2980. In this earlier work they *per*-benzylated oxazolidinethiones derived from D-arabinose, L-arabinose, D-xylose, D-ribose, D-benzylfructose and D-benzylsorbitose to the corresponding S-benzylthiones. It was Tatibouët & colleagues who synthesized the first nucleoside-like 2'-O-bridged β -purine *arabino*-nucleosides using various anthranilic acid derivatives in ethanol, i.e., compounds that are constitutionally and configurationally very similar to the final compounds of this work. In 2004 (your ref. 32) the same group compared benzyl with silyl ethers and optimized the substitution yields with a wider range of anthranilic acid derivatives by replacing ethanol with *tert*-butanol. The ring closure to give quinazolinones required the elimination of water from anthranilic acid derivatives by heat, and so they used absolute alcohol solvents in the presence of molecular sieves. Therefore, to write that this substitution on S-benzylthiones was "limited to ... organic solvents" misses a point. The reports are limited to anthranilic acid derivatives in ethanol and *tert*-butanol (out of all organic solvents). It may well be that anthranilic nucleosides would form in water (or water-alcohol mixtures) prior to the ring closure to quinazolinone nucleosides. But it was not their goal to test this. So this sentence perhaps deserves slight re-phrasing as well.

"We hypothesized that thione protonation and weak amine solvation were both essential to these limited examples." It is true that your secret lies in weakly acidic conditions that do not fully protonate **2b** and **2c**, thus blocking the nucleophilicity of the amino components, but still protonate the thione. This is difficult to imagine. So this reviewer asks: What is the pK_a of protonation of thione **1c**? Are we talking about sulfur protonation, which indeed would give C2 a marked carbocationic character, or is there tautomerization from the nitrogen involved? These questions appear while reading the manuscript several times. Also, what is the pK_a of deprotonation of this thione? Furthermore, what are the pK_a values of protonation of S-methylthiones as first synthesized by Coxon & colleagues in ref. 31? What are the pK_a values of protonation of S-benzylthiones as prepared by Tatibouët & colleagues (in ref. 32)? (Replace at least three times alkyl thione with S-alkyl thione, please.)

First line on page 8: Supplementary Fig. 12. Besides the typo in the last word of the legend, how is MeS-derivative *arabino*-**1c** (the magenta doublet in spectrum B at about 6.15 ppm) distinguished from the (green) doublet resonating at approximately the same frequency, a signal that was attributed to the

anthranil intermediate *arabino-10e* in the lower spectrum A? In other words, are you sure that at pH 8 virtually nothing happens ("severely retarded")? Only then you could call this a "hypothesised pH-switch" (page 7, line minus 5). Is it possible that at pH 8 this doublet at 6.15 ppm should be stained green, thus indicating appreciable amounts of anthranil intermediate *arabino-10e*? This reviewer is astonished about the high mobility of the green doublet through the analyzed pH range. Could you comment on this please? Given that the argument of "thione protonation" has an important bearing in this work, another spiking experiment at pH 8 with synthetic (green) **10e** (in B) and/or, the other way around, with synthetic (magenta) **1c** (in A) would help to eliminate this doubt.

In the middle of page 8 the authors write: "... incubation of *arabino-1b* or *ribo-1b* (0.25 M) with ammonium chloride (1 M, pH 8.5 – 10.5) returns arabino- and riboaminooxazoline **10a** (15 – 23%), from their respective thiones **1b**. Interestingly, the major by-products are the precursor thione **1a** (37 – 42%) and a white crystalline precipitate of dicyanovinylsulfide **15** (Figure 3)³³; these by-products suggested regeneration of thione **1a** results from rapid nucleophilic addition of thiolate **13** to the cyanovinyl moiety of **1b**." This expression reads overly anthropocentric because, if thione **1a**, obtained in 37-42 % yields, is called a by-product, are then *arabino*- and *ribo*aminooxazolines **10a**, that were obtained in 15-23 % yields, seen as the main products? Shouldn't this be objectively the other way around? For most chemists the main product is the one obtained with the highest molar yield, rather than the most desired compound.

Page 9. Purine Elaboration: "The oligomerisation of hydrogen cyanide has widely been proposed as a key route to purine nucleobases,^{3,7-10} and ..." This is not quite correct in the same sense as mentioned in the introduction. Refs. 3, 8, 9 and 10 do indeed propose HCN oligomerization as a key route, but Becker et al. in ref. 7 just mention, comment and cite (some of) these propositions while actually proposing an alternative pathway, irrespective of its prebiotic plausibility all the way through.

Page 10: "It is likely that the combination of thione protonation and the remarkably low amine pK_a (decreased through the inductive effect of the nitrile moiety)⁴¹ results in the now facile displacement of sulfide from alkyl thione **1c**." Here two pK_a values are needed, even if they were just calculated estimates.

Last two sentences on page 10: "The improved efficiency of dinitrile cyclisation is likely due to both nitrile-nucleophile proximity and the increase electron withdrawal of the α -nitrile with respect to the α -amide. It is also noted that though reaction of **1c** with glycine nitrile (**2d**) yielded aminooxazoline **10d** in excellent (76%) yield, cyclisation of **10d** was not observed under any condition investigated and was readily isolated as the uncyclised aminooxazoline (61%)." What is described as proximity is not quite clear, since the closest proximity of the cyano group in the attached glycine nitrile **10d** is the same as the closest proximity in the attached malononitrile or caboxamido nitriles **10b** and **10c**. However, the probability of the intramolecular nucleophile hitting the nitrile is twice as large in **10b** than in **10c** or **10d**. One should expect to observe an entropic (statistical) effect similar to the chelate effect. In addition, how true and important is electron withdrawal of nitriles with respect to that of carboxamides? Which functional group is more readily hydrolyzed, a nitrile or the corresponding amide? Nitriles are quite inert (triple-bonds) when it comes to nucleophilic attack on an unprotonated nitrile. On the other hand, in the glycine nitrile **10d** being, as the authors correctly emphasize, at a more reduced oxidation level than both **10b** and **10c**, there is no conjugation, thus, a much weaker rotational barrier. The cyano group has more available space to move away from the nucleophilic β -nitrogen atom, hence, this is again an entropic effect. Maybe an isolated nitrile (like in glycine nitrile **10d**) is in principle (i.e. in an intermolecular reaction) even more electrophilic than a conjugated nitrile (like in **10b** and **10c**)? Given that the lack of ring closure is important for the conclusion of the whole work, this part should be rewritten.

Page 11, middle: "... wobble base-pairing inosine ..." Inosine makes wobble base pairs with uridine or thymidine. With cytidine it pairs in Watson-Crick mode, if only with two H-bonds, thus, less stably than GC. Please refer to, for example, *Curr. Opin. Genet. Develop.* **2014**, *26*, 116-123 or some other work that describes the stability of IC versus IU in RNA. If inosine can base-pair with cytidine in RNA, it could have evolved as a guanosine substitute (at least at low temperatures).

First four lines on page 13: "... incubation of **30A/180A** (22% + 33%) with either glycerol (10 equiv.) or cytidine (1 equiv.) leads to an increased yield of cyclic phosphate **30A** (38% and 41%, respectively), demonstrating the predisposition of cyclic phosphate synthesis during urea-mediated phosphorylation." In the supplementary file Methods I-V are described, where Method I is the urea-mediated phosphorylation in the absence of adducts, II is the urea-mediated phosphorylation with added formamide, III with glycerol, IV with cytidine and V with ancitabine (**11**). A supplementary Table 2 summarizes the results. Only a small part of it is mentioned in the main text, why? Why would the presence of cytidine change the outcome? C and **11** are in principle complementary to **17A** not **17I**, but not at 140 °C, viz. not at the phosphorylation (incubation) temperature applied here. So base-pairing could not have been the criterion. The most evident

effect of all the additives, including glycol and formamidine, is the strong diminution of the 5'-phosphate with respect to the 2',3'-cyclic phosphate contents, which is not at all mentioned in the main text. The sole mentioning of increased yields of 2',3'-cyclic phosphates when using Methods III and IV appears thus somewhat arbitrary. This reviewer finds it regrettable not to read more about these experiments at the end of this section in the main text.

Page 13. Chemical Divergence of Purines and Pyrimidines: "... allowing for the concomitant synthesis of pyrimidine and purine anhydronucleotides **11** and **16c** through ..." Strictly, whereas **11** is indeed a pyrimidine anhydronucleotide, **16c** is not a purine anhydronucleoside. This compact sentence ignores the main lack of this work, namely, the fact that the final ring closure to a true purine anhydronucleoside through aminoimidazole formylation performed (best) with formamidine has only been described in separate experiments, not in the presence of **11**. What is there the reason for this? This question is likely to be asked by other readers. It should be explicitly addressed in the conclusion. The above sentence should be modified to say that precursors of purine anhydronucleosides can be synthesized concomitantly with pyrimidine anhydronucleotides.

Page 14, last sentence before the Conclusion: "... thereby facilitating access to the RNA world." Strictly: ... facilitating access to the prebiotic synthesis of RNA. The RNA world rests an unproven hypothesis, it implies far more than just the presence of prebiotic RNA. This reviewer recommends to withhold from the RNA world and just stay with the experimental facts.

Conclusion: "These highly selective cyclisations strongly suggest that renewed investigation of prebiotic aminonitrile **2b** is warranted, avoiding the uncontrolled high-pH oligomerisation of hydrogen cyanide." It would seem appropriate to cite ref. 7 after "warranted", since **2b** is central to this other approach published by Becker et al.

Page 15, first sentence: "... pyrimidine and 8-oxo-purine ribonucleotide-2',3'-cyclic phosphate **30A** and **30I** syntheses, **30A** and **30I** are good candidates for monomeric units in the early stage of replication and template directed RNA synthesis." One concern is not addressed here, the (Lewis) acid enhanced depurination tendency of 8-oxopurines in RNA. In this particular context, the available data on 8-oxoA and 8-oxoG depurination as nucleosides and in DNA or RNA should be mentioned.

Typo in Figure 2 legend: alkyl-sulfide (missing I)

Proposition for Figure 1 in the main paper: Graphically, at first sight, one has the impression that ammonolysis using **2a** works on all oxazolidinethiones **1a**, **1b** and **1c**, but this is not true (as explained in the text). To avoid this impression, I would suggest to draw two explicit arrows (to the left) from **1a** and **1b** showing that NH_3 **2a** leads to aminooxazolidine **10a**, and a third, clearly crossed arrow (with NH_3 **2a**) indicating that S-methylthiones do not react with ammonia to give **10a**.

Response to reviewers' comments:

Reviewer #1

We thank reviewer #1 for their insightful comments and thorough appraisal of our manuscript.

Comment 1: The letter starts quite bold with that “Understanding prebiotic nucleotide synthesis is one of the most important challenges in elucidating the origins of life on Earth.”. Well, honestly, I am not a big fan that papers have to state that its own mono-disciplinary viewpoint is the most important one - especially in this field where many equally important challenges have to be solved to reconstruct the origin of life. I think this could be tuned down.

Response 1: We have modified this text to begin as follows: *Prebiotic nucleotide synthesis is a long standing challenge thought to be essential to elucidating the origins of life on Earth.*

Comment 2: Some more text in Figure 1 and a longer Figure caption could help in guiding the reader.

Response 2: We have added more text to Fig. 2 caption.

Comment 3: Perhaps it would be good to add a Figure 0 with the main message of the paper and link this to abstract and introduction.

Response 3: We have introduced an “overview” figure (Fig. 1).

Comment 4: Perhaps it is useful to visually highlight the divergence in Figure 1 so that also (1) is found faster on Figure 1.

Response 4: We have provided a graphical indication of the points of divergence and convergence during nucleotide synthesis in Fig. 2.

Comment 5: While biological results on DNA are used to motivate the intermediate purines and many references link to DNA, there is no discussion on how this scheme would be compatible with a co-synthesis of both RNA and DNA, something that would be not futile even in an RNA world scenario. Perhaps the authors want to comment on this?

Response 5: We thank reviewer #1 for these insightful comments and suggestions. The motivation to study 8-oxo-purines has previously been largely driven by investigating DNA oxidation, therefore much of the literature about 8-oxo-nucleotides focuses on DNA rather than RNA. We have previously explored the congruent chemical origins of RNA and DNA (see ref. 13), and we agree a congruent synthesis of RNA and DNA would be highly appealing in the context of the origins of life, however exploring the concomitant synthesis of DNA and RNA is beyond the scope of our current manuscript.

Reviewer #2

We thank reviewer #2 for the time and care they have taken in reviewing our manuscript. We are most grateful for their comments and suggested improvements to our manuscript.

Comment 1: Please be more precise and thus more fair: Which steps were low yielding, which ones were implausibly prebiotic and the spectrum of which isomeric compounds was wide?

Response 1: We apologise if our brevity did any injustice to the progress made by Becker *et al* in ref. 7.

We have provided a more specific analysis of the results reported:

“A notable proposed pathway to the purine nucleotides, that achieves excellent N9-selectivity during ribosylation,⁷ remains problematic due to an unselective ribosylation (that furnished a mixture of natural furanosyl and non-natural pyranosyl isomers, and a mixture of natural b-anomers and non-natural a-anomers). Furthermore, this strategy, a variant of the classic Traube purine synthesis,⁸ generates a wide spectrum of glycone isomers, homologues and anomers alongside a low yield of natural nucleosides from prebiotically plausible sugar mixtures.⁷”

We have included a reference that provides an overview of Traube purine synthesis strategy.

[8] Wang, Z. Traube purine synthesis. *Comprehensive organic name reactions and reagents*. **625**, 2789–2792 (2010).

Comment 2: "Remarkably, all proposed prebiotic syntheses of pyrimidine and purine nucleotides are mutually incompatible;⁴⁻¹⁰". On what scientific, not intuitive, grounds can the authors conclude that all these syntheses are mutually incompatible?

Response 2: We have modified a) the main text and b) the abstract:

a)
“Remarkably, all proposed prebiotic syntheses of pyrimidine and purine nucleotides have yielded either pyrimidines or purines separately, but never (yet) both by the same strategy,^{4-7,9-11} whilst purine nucleotides can be synthesised by direct ribosylation, pyrimidine nucleotides cannot, and though a complete stepwise pyrimidine nucleotide synthesis has been demonstrated (even from complex sugar mixtures⁵) a comparable strategy for purine synthesis remains elusive.⁴⁻⁶”

b)
“no divergent synthesis from common precursors have been proposed. Moreover, the prebiotic syntheses of pyrimidine and purine nucleotides that have been demonstrated operate under mutually incompatible conditions.”

We note that the pathways described to access canonical pyrimidine nucleotides (ref. 4-6) require access to anhydronucleotides (e.g. compound **11**). Accordingly low to neutral pH (pH ≤ 6.5) reactions conditions are absolutely required to preserve the 2,2'-anhydronucleotide bond.

Conversely, all reported purination reactions require large pH fluctuations, with initial dry state heating under acid condition followed by a highly alkaline subsequent phase, employing either hydroxide, ammonia, borax or triethylamine solutions (see ref. 7 & 10).

These two sets of reaction conditions are (currently) mutually incompatible.

Comment 3: Neither on page 5, in the schemes or in the supplemental file have I found the information on whether racemic or enantiopure glyceraldehyde (**7**) was used and, if enantiopure, which, D or L?

Response 3: We have modified the supplementary information to indicate that all three forms of glyceraldehyde have been investigated. Racemic glyceraldehyde, homochiral D-glyceraldehyde and homochiral L-glyceraldehyde are all commercially available and have all been successfully employed in this reaction.

The chirality of glyceraldehyde, as expected, has no effect on the reaction yield or diastereoselectivity and glyceraldehyde's chirality is relayed from starting material to product.

Though we recognize it is an extremely interesting question, understanding the prebiotic availability of homochiral (or scalemic) glyceraldehyde to yield a chiroselective/chiral pool prebiotic synthesis of purine

nucleotides is beyond the scope of our current manuscript.

Comment 4: "Here, we envisaged cyanoacetylene (**8**) as ideally suited to activate thione **1a** in aqueous solution due the slow reaction of cyanoacetylene (**8**) with water and the excellent electron withdrawing properties of the cyanovinyl moiety.^{28,29}. (-> moiety) At this stage it is not clear why excellent electron withdrawing properties would be important.

Response 4: Following "thione activation" the next step of our synthesis is thiol displacement by nucleophilic attack at C2 (as shown in Fig. 2). Therefore a strongly electron withdrawing cyanovinyl group will activate the C2-carbon atom to nucleophilic attack.

We have modified the text:

*"... and the excellent electron withdrawing properties of the cyanovinyl moiety, that would activate the C2 carbon atom of **1** to nucleophilic addition."*

Comment 5: "Due to the higher pKa of S-cyanovinyl thione **1b** and sulfur-prohibited annulation, no increase... " Higher pKa than pKa of what?

Response 5: We have modified the text:

*"Due to the higher pK_a of anhydronucleotide **11** than S-cyanovinyl thione **1b**,³¹ and sulfur-prohibited annulation, no increase in pH was observed during cyanovinylation of thione **1a**. Increasing pH is a hallmark of the addition of cyanoacetylene (**8**) to aminooxazoline (**10a**) in water, rendering pH-buffered cyanovinylation essential to pyrimidine synthesis, however no buffer was required to control the reaction of **1a** with cyanoacetylene (**8**)."*

We have added reference 31, which provides measurements of compound **11** pK_a's.

[31] Walwich, E. R., Roberts, W. K. & Dekker, C. A. Cyclisation during the phosphorylation of uridine and cytidine by polyphosphoric acid: A new route to the O²,2'-cyclonucleosides. *Proc. Chem. Soc.* 84 (1959).

Comment 6: "... to our knowledge, thiolate displacement from **1c** (R = Me) by ammonia (**2a**) to yield **10a** has only been observed upon reflux in "undistilled DMF" (which was likely contaminated with ammonia (**2a**) and formic acid).³¹" Careful, dimethylformamide does not produce ammonia and formic acid. In ref. 31 the reactants were heated for 3 hours at 90 °C (not reflux) in "technical grade formamide" (not DMF), which was likely contaminated with ammonium formate that can slowly release ammonia and formic acid.

Response 6: We have amended the text as follows to correct our error (DMF should have been formamide):

*"thiolate displacement from **1c** (R = Me) by ammonia (**2a**) to yield **10a** has only been observed upon "treatment with formamide at 90°C for 3 h" (which was likely contaminated with ammonium formate and can slowly release ammonia (**2a**) and formic acid)."*

Comment 7: Please cite (in addition to or replace with) Girniene, J.; Gueyrard, D.; Tatibouët, A.; Rollin, P. *Tetrahedron* 2001, 42, 2977-2980.

Response 7: We had previously cited this paper in the supplementary information (see SI ref. 3). We have now also cited this paper in the manuscript:

Girniene, J., Gueyrard, D., Tatibouët, A., Sackus, A. & Rollin, P. Base-modified nucleosides from carbohydrate derived oxazolidinethiones: a five-step process. *Tetrahedron* **42**, 2977-2980 (2001).

Comment 8: It was Tatibouët & colleagues who synthesized the first nucleoside-like 2'-O-bridged β -purine arabino-nucleosides using various anthranilic acid derivatives in ethanol, i.e., compounds that are constitutionally and configurationally very similar to the final compounds of this work.

Response 8: Tatibouët & colleagues synthesised pseudo-uridine analogues with a [5,5,6,6] ring structure.

We on the other hand synthesized "2'-O-bridged β -purine arabino-nucleosides" containing a different fused ring structure. Our key ring structure contains a fused [5,5,5,6] ring structure. It is our [5,5,5,6] structure that is key to 8-oxo-purine synthesis.

C2'-inversion (or C2-hydrolysis) of Tatibouët's [5,5,6,6] structures leads to pseudo-uridines not 8-oxo-purines.

Comment 9: The reports are limited to anthranilic acid derivatives in ethanol and tert-butanol (out of all organic solvents).

Response 9: We have now specified the organic solvents used by Tatibouët and co-workers, rather than using the general term "organic solvents".

Comment 10: What is the pKa of protonation of thione **1c**? Are we talking about sulfur protonation, which indeed would give C2 a marked carbocationic character, or is there tautomerization from the nitrogen involved? These questions appear while reading the manuscript several times. Also, what is the pKa of deprotonation of this thione? Furthermore, what are the pKa values of protonation of Smethylthiones as first synthesized by Coxon & colleagues in ref. 31? What are the pKa values of protonation of S-benzylthiones as prepared by Tatibouët & colleagues (in ref. 32)?

Response 10: We have provided the pKa of thione **1c** in water.

We have not made Tatibouët's tetrabenzylthione (**1d**) or their silyl-protect analogues. We do not anticipate these are water soluble or prebiotically plausible. They do not form part of our investigation.

These reactions are all carried out in water and the molecules are surrounded by a hydration sphere, but protonation at N1 is expected and provides considerable carbocation character at C2, as shown below by the resonance forms of the protonated thione.

Comment 11: (Replace at least three times alkyl thione with S-alkyl thione, please.)

Response 11: We have corrected all instances, as requested.

Comment 12: First line on page 8: Supplementary Fig. 12. Besides the typo in the last word of the legend, how is MeS derivative arabino-1c (the magenta doublet in spectrum B at about 6.15 ppm) distinguished from the (green) doublet resonating at approximately the same frequency, a signal that was attributed to the anthranil intermediate arabino-10e in the lower spectrum A? In other words, are you sure that at pH 8 virtually nothing happens ("severely retarded")?

Response 12: Yes. As shown, "virtually nothing happened" over this period of incubation at pH 8. However the reaction does sluggishly proceed at pH 8, as can be seen in the spectra provided. We have

supplied a specific time point for direct comparison of each pH. We recognized the reaction was not completely prevented, and therefore used the term “severely retarded”, given the large observed difference between pH 6 and pH 8.

We have now added further spectra to this figure in the supplementary information, giving a later time point at pH 8 in addition to the spectra provided in Supplementary Fig. 12 (now Supplementary Fig. 17). This time point was adjusted to pH 4 so that direct comparison between chemical shifts of the (C1')—H resonances of the products formed at pH 4 and pH 8. This demonstrates the clear difference between the green and magenta compounds that we have assigned.

In the additional figure we have also provide a wider chemical shift window and clearly marked the (C2')—H that is highly characteristic in splitting, J value and chemical shift for these two compounds, and can be used to assign magenta accurately at pH 8.

Furthermore we have provided a spike of the magenta compound to unambiguously demonstrate our correct assignment in the pH spectra.

Comment 13: This reviewer is astonished about the high mobility of the green doublet through the analyzed pH range. Could you comment on this please?

Response 13: The pKa of an aminooxazoline is 6.5. When this pH (pH = pka) is crossed on moving from pH 8 to 4 the protonation state of the aminooxazoline changes dramatically (according to the Henderson-Hasselbalch equation). Consequently, the ¹H-NMR chemical shift of (C1')—H also changes dramatically. This protonation induced deshielding of (C1')—H is not unusual.

Comment 14: Given that the argument of "thione protonation" has an important bearing in this work, another spiking experiment at pH 8 with synthetic (green) 10e (in B) and/or, the other way around, with synthetic (magenta) 1c (in A) would help to eliminate this doubt.

Response 14: We have provide this additional spiking experiment (see Response 12)

Comment 15: "... incubation of arabino-1b or ribo-1b (0.25 M) with ammonium chloride (1 M, pH 8.5 – 10.5) returns arabino- and riboaminooxazoline 10a (15 – 23%), from their respective thiones 1b. Interestingly, the major by-products are the precursor thione 1a (37 – 42%) and a white crystalline precipitate of dicyanovinylsulfide 15 (Figure 3)³³; these by-products suggested regeneration of thione 1a results from rapid nucleophilic addition of thiolate 13 to the cyanovinyl moiety of 1b." This expression reads overly anthropocentric because, if thione 1a, obtained in 37-42 % yields, is called a by-product, are then arabino- and riboaminooxazolines 10a, that were obtained in 15-23 % yields, seen as the main products? Shouldn't this be objectively the other way around? For most chemists the main product is the one obtained with the highest molar yield, rather than the most desired compound.

Response 15: A by-product is an incidental or secondary product made in the manufacture or synthesis of something else.

We state clearly our intention to synthesize aminooxazoline **10a**. Therefore all other products (irrespective of their relative yield) are by-products of this aminooxazoline synthesis.

Comment 16: Page 9. Purine Elaboration: "The oligomerisation of hydrogen cyanide has widely been proposed as a key route to purine nucleobases,^{3,7-10} and ..." This is not quite correct in the same sense as mentioned in the introduction. Refs. 3, 8, 9 and 10 do indeed propose HCN oligomerization as a key route, but Becker et al. in ref. 7 just mention, comment and cite (some of) these propositions while actually proposing an alternative pathway, irrespective of its prebiotic plausibility all the way through.

Response 16: We disagree. The pathway described by Becker *et al* is ultimately reliant upon HCN-trimers **2b** and **2c**, which are required to build their key intermediates FaPyG and FaPyA en route to

purines.

Becker *et al* specifically state: "The FaPy pathway starts with the formamidopyrimidines 11 to 13 which are prebiotically available from simple starting materials (encircled) via multi-aminopyrimidines. The molecules in red are derived directly from NH₄CN."

Comment 17: Page 10: "It is likely that the combination of thione protonation and the remarkably low amine pKa (decreased through the inductive effect of the nitrile moiety)⁴¹ results in the now facile displacement of sulfide from alkyl thione 1c." Here two pKa values are needed, even if they were just calculated estimates.

Response 17: These pKa's have been added to the manuscript and titration curves reported in the SI for all new pKa's.

Comment 18: " What is described as proximity is not quite clear.

Response 18: We have modified this text, changing "proximity" to the more accurate term "effective molarity", which compasses the statistical and entropic effects.

Comment 19: In addition, how true and important is electron withdrawal of nitriles with respect to that of carboxamides?

Response 19: The pK_a of acetonitrile is 31, whereas the pKa of diethylacetamide is 35. The pKa of malononitrile is 11, whereas the pKa of cyanoacetamide is 17. These pKa's give a clear indication of the increased anion stabilizing effect of a nitrile over an amide. Furthermore nitriles are deemed to have a "strong electron-withdraw effect", whereas amides only have a "moderate electron-withdrawing effect". Specifically, for a carbonitrile moiety $\sigma_p = 0.66$, $\sigma_m = 0.56$, whereas for a carboxamide $\sigma_p = 0.36$, $\sigma_m = 0.28$.

Hansch, C., Leo, A. & Taft, R. W. A survey of Hammett substituent constants and resonance and field parameters. *Chem. Rev.* **97**, 165-195 (1991).

Comment 20: Which functional group is more readily hydrolyzed, a nitrile or the corresponding amide? Nitriles are quite inert (triple-bonds) when it comes to nucleophilic attack on an unprotonated nitrile.

Response 20: During the synthesis of **16c** we only observed (*5-exo-dig*) cyclisation of the nitrile moiety of **10c**. No reaction with the amide moiety of **10c** is observed, indicating the chemoselectivity of reaction of the nitrile moiety (not the amide moiety).

Comment 21: Maybe an isolated nitrile (like in glycine nitrile 10d) is in principle (i.e. in an intermolecular reaction) even more electrophilic than a conjugated nitrile (like in 10b and 10c)?

Response 21: The nitriles in **10b** and **10c** are not conjugated, there is an sp³-hybridised carbon between the sp² centers.

Comment 22: Please refer to, for example, *Curr. Opinion Genet. Develop.* 2014, 26, 116-123 or some other work that describes the stability of IC versus IU in RNA.

Response 22: We have added this reference:

Alseth, I., Dalhus, B. & Bjørås, M. Inosine in DNA and RNA. *Curr. Opinion Genet. Develop.* **26**, 116–123 (2014).

Comment 23: Why would the presence of cytidine change the outcome? C and 11 are in principle complementary to 17A not 17I, but not at 140 °C, viz. not at the phosphorylation (incubation) temperature applied here. So base-pairing could not have been the criterion.

Response 23: The formation of cyclic phosphodiester is irreversible under these urea-mediated phosphorylation conditions, therefore the addition of a vicinal diol (such as glycerol or cytidine) will reverse the 5'-monophosphate and sequester phosphate from bis-phosphate **18OA**, this can be used to improve the total yield of **3OA**. The restricted rotation about the vicinal diol of cytidine makes cytidine much more effective than glycerol in achieving this goal and therefore fewer equivalents were employed.

We have modified the text to clearly outline the role of both glycerol and cytidine as diols in these reactions.

“due to the irreversible nature of 2',3'-cyclic phosphate synthesis, further incubation of 3OA/18OA (22% + 33%) with a diol, for example glycerol (10 equiv.) or cytidine (1 equiv.), leads to an increased yield of cyclic phosphate 3OA (38% and 41%, respectively) by sequestering phosphate from the (reversible) equilibrium with the 5'-phosphate moiety of 18OA (Supplementary Table 2)”

Comment 24: "... allowing for the concomitant synthesis of pyrimidine and purine anhydronucleotides 11 and 16c through ..." Strictly, whereas 11 is indeed a pyrimidine anhydronucleotide, 16c is not a purine anhydronucleoside. This compact sentence ignores the main lack of this work, namely, the fact that the final ring closure to a true purine anhydronucleoside through aminoimidazole formylation performed (best) with formamide has only been described in separate experiments, not in the presence of 11. What is the reason for this? This question is likely to be asked by other readers. It should be explicitly addressed in the conclusion. The above sentence should be modified to say that precursors of purine anhydronucleosides can be synthesized concomitantly with pyrimidine anhydronucleotides.

Response 24: We have added an additional reaction to the paper to demonstrate that formylation of **16c** to yield **17A** can be undertaken in the presence of **11**.

We have added two figures to the supplementary information:

a) A NMR time course for the synthesis of **17A** from **16c** in the presence of **11** in formamide.

b) NMR spectra to show this same reaction after transfer to aqueous solution with authentic standard spikes for **16c**, **17A**, **11** and ara-cytidine (the hydrolysis product of **11**)

Comment 25: Page 14, last sentence before the Conclusion: "... thereby facilitating access to the RNA world." Strictly: ... facilitating access to the prebiotic synthesis of RNA. The RNA world rests on an unproven hypothesis, it implies far more than just the presence of prebiotic RNA. This reviewer recommends to withhold from the RNA world and just stay with the experimental facts.

Response 25: The modification "RNA world" to "prebiotic synthesis of RNA" has been made.

Comment 26: "These highly selective cyclisations strongly suggest that renewed investigation of prebiotic aminonitrile 2b is warranted, avoiding the uncontrolled high-pH oligomerisation of hydrogen cyanide." It would seem appropriate to cite ref. 7 after "warranted", since 2b is central to this other approach published by Becker et al.

Response 26: This cross-reference has been added. We have also added a reference to a recent highlight/analysis of Becker et al's paper, which also indicates that "there is still work to be done to find the missing prebiotic link—or yet another approach" to prebiotic purine synthesis.

[54] Fiore, M. & Strazewski, P. Bringing prebiotic nucleosides and nucleotides down to Earth. *Angew. Chem. Int. Ed.* **55**, 13930–13933 (2016).

Comment 27: "... pyrimidine and 8-oxo-purine ribonucleotide-2',3'-cyclic phosphate 3OA and 3OI syntheses, 3OA and 3OI are good candidates for monomeric units in the early stage of replication and template directed RNA synthesis." One concern is not addressed here, the (Lewis) acid enhanced depurination tendency of 8-oxopurines in RNA. In this particular context, the available data on 8-oxoA and 8-oxoG depurination as nucleosides and in DNA or RNA should be mentioned.

Response 27: 8-oxo-dG and 8-oxo-dA are more stable than dG and dA to acid hydrolysis:

Bialkowski, K., Cysewski, P. & Olinski, R. Effect of 2'-deoxyguanosine oxidation at C8 position on N-glycosidic bond stability *Z. Naturforsch. C Bio. Sci.* **51**, 119-122 (1995).

Theruvathu, J. A., Jaruga, P., Dizdaroglu, M. & Brooks, P. J. The oxidatively induced DNA lesions 8,5'-cyclo-2'-deoxyadenosine and 8-hydroxy-2'-deoxyadenosine are strongly resistant to acid-induced hydrolysis of the glycosidic bond. *Mech. Ageing Dev.* **128**, 494-502 (2007).

Fleming, A. M., Alshykhly, O. Zhu, J., Muller, J. G. & Burrows C. J. Rates of chemical cleavage of DNA and RNA oligomers containing guanine oxidation products. *Chem. Res. Toxicol.* **28**, 1292–1300 (2015).

Ribonucleotides are roughly 500-times more stable to acid hydrolysis than deoxyribonucleotides:

Zoltewicz, J. A., Clark, D. F., Sharpless, T. W. & Grahe, G. Kinetics and mechanism of the acid-catalyzed hydrolysis of some purine nucleosides *J. Am. Chem. Soc.* **92**, 1741-1750 (1970).

There is no obvious concern with respect to the stability of the glycosidic bonds of 8-oxo-purine ribonucleotides and we saw no evidence of cleavage of the glycosidic bond of 8-oxo-purines during phosphorylation. We see no indication that the products are unstable to depurination.

We have made it clear that the glycosidic bond of 8-oxo-purine is remarkably stable.

Whether the stability of glycosidic bonds was a significant factor in the selection of biology's purine nucleotides is beyond the scope of our manuscript, however given that 8-oxo-dA and 8-oxo-dG are significantly more stable than dA or dG it is not obvious that this stability was a primary factor in the ultimate selection.

Comment 28: Typo in Figure 2 legend: alkyl-sulfide (missing l).

Response 28: Corrected.

Comment 29: Proposition for Figure 1 in the main paper: Graphically, at first sight, one has the impression that ammonolysis using 2a works on all oxazolidinethiones 1a, 1b and 1c, but this is not true (as explained in the text). To avoid this impression, I would suggest to draw two explicit arrows (to the left) from 1a and 1b showing that NH₃ 2a leads to aminooxazolidine 10a, and a third, clearly crossed arrow (with NH₃ 2a) indicating that S-methylthiones do not react with ammonia to give 10a.

Response 29: We have specified under each arrow which R-group was used.

REVIEWERS' COMMENTS:

Reviewer #1 (Remarks to the Author):

I think this is a very solid and careful resubmission. I recommend publication as is.

Reviewer #2 (Remarks to the Author):

Response to the revised version:

The Powner lab thus has set the new Gold Standard in Supplementary Information documentation.

Response to Reviews Comments

Reviewer #1:

Comment: I think this is a very solid and careful resubmission. I recommend publication as is.

Response: We thank reviewer #1 for these comments.

Reviewer #2:

Comment: Response to the revised version: The Powner lab thus has set the new Gold Standard in Supplementary Information documentation.

Response: We thank reviewer #2 for their appraisal of our Supplementary Information.